# ERK2-topoisomerase II regulatory axis is important for gene activation in immediate early genes

The function of the mitogen-activated protein kinase signaling pathway is required for the activation of immediate early genes (IEGs), including *EGR1* and *FOS*, for cell growth and proliferation. Recent studies have identified topoisomerase II (TOP2) as one of the important regulators of the transcriptional activation of IEGs. However, the mechanism underlying transcriptional regulation involving TOP2 in IEG activation has remained unknown. Here, we demonstrate that ERK2, but not ERK1, is important for IEG transcriptional activation and report a critical ELK1 binding sequence for ERK2 function at the *EGR1* gene. Our data indicate that both ERK1 and ERK2 extensively phosphorylate the C-terminal domain of TOP2B at mutual and distinctive residues. Although both ERK1 and ERK2 enhance the catalytic rate of TOP2B required to relax positive DNA supercoiling, ERK2 delays TOP2B catalysis of negative DNA supercoiling. In addition, ERK1 may relax DNA supercoiling by itself. ERK2 catalytic inhibition or knock-down interferes with transcription and deregulates TOP2B in IEGs. Furthermore, we present the first cryo-EM structure of the human cell-purified TOP2B and etoposide together with the *EGR1* transcriptional start site (−30 to +20) that has the strongest affinity to TOP2B within −423 to +332. The structure shows TOP2B-mediated breakage and dramatic bending of the DNA. Transcription is activated by etoposide, while it is inhibited by ICRF193 at *EGR1* and *FOS*, suggesting that TOP2B-mediated DNA break to favor transcriptional activation. Taken together, this study suggests that activated ERK2 phosphorylates TOP2B to regulate TOP2-DNA interactions and favor transcriptional activation in IEGs. We propose that TOP2B association, catalysis, and dissociation on its substrate DNA are important processes for regulating transcription and that ERK2-mediated TOP2B phosphorylation may be key for the catalysis and dissociation steps.

Transcription is the first step in gene expression, where RNA is synthesized by a DNA-dependent RNA polymerase. In eukaryotic cells, RNA polymerase II (Pol II) is the core enzyme for the transcription of protein-coding mRNA and some non-protein coding RNA[1]. Pol II routinely synthesizes mRNAs on house-keeping genes, whereas Pol II activity is timely and tightly regulated by transcriptional activators and repressors on inducible genes. In particular, stress-inducible genes respond to environmental signals and cellular needs, and their synchronized and rapid gene expression is crucial for cell survival, maintenance, and growth[2–6]. Representative examples are serum/growth

✉e-mail: heeyounbunch@gmail.com

factor-inducible immediate early genes (IEGs)[3,4], heat shock genes[7–9], neurotransmitter-inducible genes[10,11], enhancer RNA genes[12,13], and hypoxia-inducible genes[14,15].

Recent studies have identified key regulatory mechanisms of these stress-inducible genes in metazoan cells. To ensure synchronized and rapid transcriptional onset, Pol II is paused at the transcription start site (TSS) during the early elongation, between transcriptional initiation and productive elongation[6,16]. Pol II promoter-proximal pausing (Pol II pausing hereafter) is caused by multiple elements, namely, protein factors, nucleic acid sequences and structures, and chromatin architecture[17–22]. Likewise, the release of Pol II pausing involves the collaboration of such elements as transcriptional activators, structural/topological changes of DNA, and modifications of chromatin dynamics[4,8,10,21,23–26]. For example, heat shock and serum induction activate transcriptional activators, HSF1 and MAP kinases such as ERK1 and ERK2, respectively[8,27,28]. While HSF1 is the sole master activator that binds to target gene promoters to induce transcriptional activation[29,30], ERK1 and ERK2 require upstream kinase cascades and downstream substrates, which are gene-specific, involving nuclear DNA binding transcriptional activators, such as FOS and ELK, to activate their target genes[31,32]. Previous studies showed that Pol II pausing at heat shock and IEGs is reversed by the binding of these gene specific transcriptional activators[4,21,24,33]. Simultaneous with the activator binding to its target gene promoter, the catalytic function of TOP2 is required for Pol II pause release[4,10,25,34–36]. Furthermore, transcriptional activation triggers and requires DNA damage response (DDR) signaling which is reduced when TOP2 is inhibited by ICRF193, a catalytic inhibitor of TOP2[4]. This finding suggests that the topological alteration of DNA by TOP2 is important for Pol II pause release. Consistently, other studies have reported that DNA breaks are abundant in Pol II pausing sites[37], that the TOP2 covalent cleavage complex is frequently formed in estrogen-induced transcription[38,39], and that both positive and negative supercoiling of DNA could inhibit Pol II forward-translocation[40,41]. These findings suggest that the DNA at Pol II pausing sites requires the catalytic function of TOP2 and is susceptible to DNA damage.

Human TOP2 proteins, including the two isomers TOP2A and TOP2B, resolve topological stresses during DNA metabolisms such as replication, transcription, and chromosomal compaction[42–45]. TOP2 catalyzes both positive and negative supercoiling as well as decatenation[42]. Previously, TOP2A was considered to be involved in replication while TOP2B in transcription[4,34,43,46]. However, recent studies have suggested that both TOP2A and TOP2B are important regulators of stress-inducible gene transcription[4,10,34,36,47,48]. Moreover, most conventional TOP2 chemical inhibitors, including ICRF193 and etoposide, target both enzymes, making it difficult to distinguish their effects on TOP2A and TOP2B[49]. TOP2A and TOP2B share 67% identity at the protein level, which is largely distinctive in the C-terminal domain (CTD)[50]. In complexes containing 20–30 nt double stranded DNA, the cryo-electron microscopy (cryo-EM) structure of a full-length human TOP2A was recently reported, whereas only a partial human TOP2B (445–1201 aa) lacking both terminal domains could be resolved by crystallography:[51–53] No cryo-EM structures for TOP2B have yet been reported. In the TOP2A study, the CTD (1191–1531 aa) of the enzyme was unstructured and could not be arrayed either by 2D or 3D reconstruction[51]. However, biochemical analyses with mutant TOP2A lacking the CTD suggested that the CTD allosterically regulates the catalytic activity of the enzyme[51].

Over 200 protein-coding genes can be simultaneously turned on within minutes of serum/growth factor induction. Some of these genes are potent transcription factors that activate the expression of a large number of genes for cell cycle progression[4,54,55]. Representative such IEGs include *EGR1*, *FOS*, *JUN*, and *MYC*[4,55]. ERK1 and ERK2 proteins are key MAP kinases that activate these genes by phosphorylating the downstream DNA-binding transcriptional activators including ELK1,

FOS, and MYC[3,56]. Both *EGR1* and *FOS* gene expression is also reportedly activated by ERKs and ELK1[55,56]. ERK1 and ERK2 proteins are approximately 82.6% identical, with their distinctive N-terminal sequences. While some studies have suggested redundant functions of ERK1 and ERK2, others have reported differential and opposite functions of ERK1 and ERK2[57–61]. Furthermore, ERK2-mediated phosphorylation of TOP2A was previously reported[62], suggesting potential functional interactions between ERK proteins and TOP2.

In this study, we attempted to understand transcriptional regulation in IEGs by ERK1 and ERK2 proteins and investigated their interaction with TOP2B. Biochemical, structural, and cell-based analyses revealed differential functions of ERK1 and ERK2 in IEG transcription and demonstrated that the CTD of human TOP2B is phosphorylated by both ERK1 and ERK2. However, ERK2, but not ERK1, activates transcription. ERK1 and ERK2 increase the rate of TOP2B catalysis to relax positive DNA supercoiling, whereas ERK2 delays the catalytic relaxation of negative DNA supercoiling by TOP2B in vitro. Our biochemical data also suggest that ERK1 may be capable of relaxing both positive and negative DNA supercoiling. The catalytic inhibition or knock-down (KD) of ERK2 caused an abnormal increase of TOP2B and repressed transcription at the representative IEGs. In addition, we present the cryo-EM structure of the human TOP2B in a ternary complex with a 50 nt segment of the *EGR1* TSS, which has a strong affinity to TOP2B, and etoposide. Etoposide treatment induces gene activation, whereas ICRF193 suppresses it at *EGR1* and *FOS*. These data visualize the interaction between TOP2B and the *EGR1* TSS and suggest a mechanism by which the DNA strand break catalyzed by TOP2B activates transcription.

## Results

### ERK2, but not ERK1, activates *EGR1* transcription

We investigated the function of ERK1 and ERK2 in IEG transcription using *EGR1* as a model gene. Previously, the *EGR1* template DNA including −432 to +332 was constructed and validated for in vitro biochemical analyses[47]. A schematic representation of the immobilized template assay is presented in Fig. 1a. The biotinylated *EGR1* template was conjugated with the avidin-coated magnetic beads. HeLa nuclear extract (NE) was used to assemble the preinitiation complex on the DNA in the presence of competitor oligomers. Recombinant ERK1 or ERK2 purified from bacteria were included along with NE (Fig. 1b; K1 for ERK1 and K2 for ERK2). Following incubation, unbound or loosely bound proteins were washed off. The proteins of interest that were stably associated with the template were detected using immunoblotting (Fig. 1a). ERK1 and ERK2 supplemented reactions were compared with an ERK storage buffer-only control (CTRL) for their ability to recruit signature factors that are important for transcriptional activation, namely, Pol II, CDK9 (a catalytic subunit of P-TEFb), and MED23 (a subunit of the Mediator complex)(Fig. 1c). Interestingly, only ERK2 consistently and notably increased all three factors, whereas ERK1 did not (Fig. 1c; Supplementary Fig. 1A).

We also considered that the effects of ERK1 and ERK2 could be compromised since the protein source of HeLa NE used for the immobilized template assay included mostly nuclear proteins, lacking the upstream, cytosolic components of the MAPK pathway needed to activate them[3,56]. Therefore, constitutively active recombinant ERK1 and ERK2 mutants, R84S and R67S[63], respectively, were purified from bacteria (Fig. 1d; ERK1 mutant as K1m/ERK1m and ERK2 mutant as K2m/ERK2m). Immobilized template assays using these proteins, comparing WT and mutant ERKs, indicated that ERK2 and ERK2m were more proficient in recruiting MED23 and CDK9 than ERK1 and ERK1m (Fig. 1e, f; Supplementary Fig. 1B). It is noted that endogenous ERK1 and ERK2 proteins in NE shown in INPUT seemed less efficient in associating with the *EGR1* template (see CTRL in immunoblotting), as if they were blocked, compared to the supplemented ERK proteins (Fig. 1e). Immobilized template assays followed by in vitro

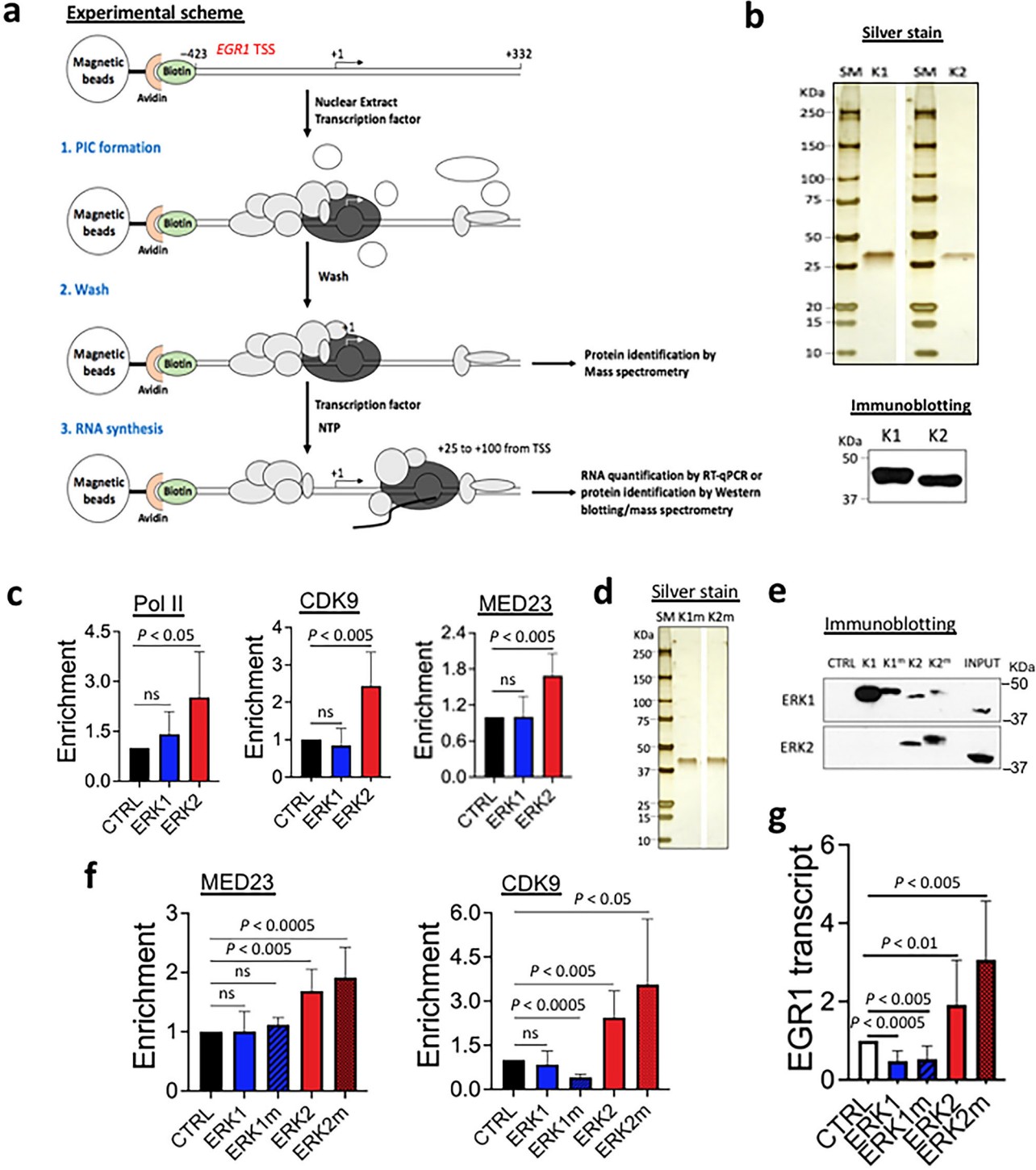

**Fig. 1 | ERK2, not ERK1, activates *EGR1* transcription. a** A schematic representation of in vitro biochemical analyses used in this study. Gray and white circles without labels, proteins; a black curved line, a nascent RNA molecule. **b** Left, silver-stained recombinant ERK1 (K1) and ERK2 (K2) proteins used in in vitro analyses. SM, standard protein size marker. Right, the result of immunoblotting confirming ERK1 and ERK2. **c** Immobilized template assay results showing the increased occupancies of Pol II, CDK9, and MED23 by ERK2 on the *EGR1* TSS. CTRL, ERK-free, buffer only control throughout the figures. Data are presented as mean values and standard deviation (SD) (*n* = 3 independent experiments*). P* values for the bar graphs were calculated with the unpaired, one sided Student's *t* test. Not significant (ns). **d** SDS-PAGE and silver-staining of the constitutively active ERK1m and ERK2m proteins used in this study. **e** Immobilized template assay showing ERK1 and ERK2 proteins associated with the template. K1, ERK1; K1ᵐ, ERK1m; K2, ERK2; K2ᵐ, ERK2m. INPUT, 20% NE used for the assay. **f** Immobilized template assay results of CDK9, and MED23 (*n* = 3 independent experiments). Data are presented as mean values and SD. *P*-values for the bar graphs were calculated with the unpaired, one sided Student's *t* test. **g** in vitro transcription assay results showing *EGR1* gene activation by ERK2 and ERK2m (*n* = 3 independent experiments). Data are presented as mean values and SD. *P* values for the bar graphs were calculated with the unpaired, one sided Student's *t* test. Source data are provided as a Source Data file.

transcription assays consistently demonstrated that ERK2 and ERK2m activated the *EGR1* transcription (Fig. 1g). In contrast, ERK1 and ERK1m repressed transcription (Fig. 1g), suggesting opposite regulatory effects of ERK1 and ERK2 on *EGR1* transcription.

### ERK2-mediated transcriptional activation requires ELK1 at the *EGR1* gene

Next, we investigated whether ERK2-mediated *EGR1* gene activation was dependent on a DNA-binding transcription factor ELK1. A previous study reported the consensus ELK1 binding sequence (CCGGAAGT) in human breast epithelial MCF10A cells using ChIP-seq[64] (Fig. 2a). The *EGR1* promoter (−423 to −1) includes numerous sites with over 75% identity to this ELK1 consensus sequence, although only one such site (GCTTCCGG, −338 to −345) has over 85% identity (87.5%; Fig. 2a). Therefore, we hypothesized that this may be a critical ELK1 binding site and to mutate this region to TAAATTAA in the immobilized *EGR1* template DNA (ELK1mut; Fig. 2a). The immobilized template assay comparing the WT and ELK1mut indicated that ELK1 binding was markedly reduced in the ELK1mut *EGR1* template (Fig. 2b, c). In addition, TOP2B and CDK9 recruitment were significantly reduced on the template (Fig. 2b, c; Supplementary Fig. 1C). We tested the transcriptional activity of WT ERKs, ERK1m, and ERK2m proteins on the WT and ELK1mut templates, using the immobilized template assay. The results demonstrated that Pol II and MED23 recruitment by ERK2 and ERK2m on the WT template was abolished to the background level on the ELK1mut template (Fig. 2d; Supplementary Fig. 1C). These data were

consistent with those from the in vitro transcription assays, showing that the ERK2-mediated transcriptional activation was strongly suppressed on the ELK1mut template (Fig. 2e). Collectively, the mutated segment, −338 to −345, appeared to be important for both ELK1 binding and ERK2 transactivation at *EGR1*, in vitro (Figs. 1c–g and 2b–e).

### ERK1 and ERK2 phosphorylate TOP2B on mutual and distinctive sites

We noted that TOP2B association with the *EGR1* TSS was positively correlated with ELK1 binding (Fig. 2b, c; Supplementary Fig. 1C). In addition, it has been previously shown that TOP2B is recruited to IEGs, including the *EGR1* gene, upon transcriptional activation[4,47]. Moreover, TOP2B inhibition using ICRF193 suppresses Pol II pause release and productive transcription in these genes[4]. Therefore, we attempted to determine the relationship between transcriptional activation and TOP2B at the *EGR1* gene. The results indicated that both ERK1m and ERK2m enhanced the recruitment of TOP2B to the *EGR1* TSS in the immobilized template assay (Fig. 3a; Supplementary Fig. 1B). However, ERK2m was more effective in recruiting TOP2B (Fig. 3a; Supplementary Fig. 1B, C). Both ERK1m and ERK2m failed to recruit TOP2B to the ELK1mut template (Fig. 3a; Supplementary Fig. 1C), suggesting that ERK-mediated TOP2B enrichment could be ELK1-dependent.

We next hypothesized that ERKs might recruit and phosphorylate TOP2B. This appears to be plausible as it was reported previously that TOP2A is phosphorylated by ERK2 in vitro[62]. To verify this hypothesis, in vitro kinase assays with human TOP2B (180 KDa) purified from

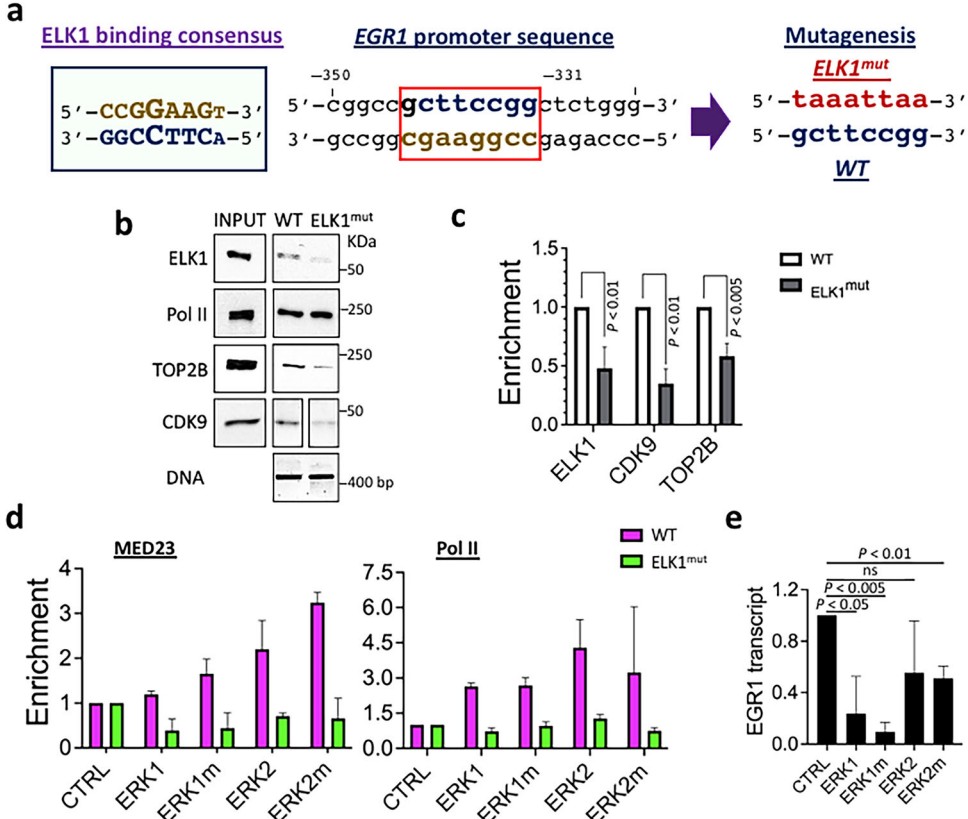

**Fig. 2 | ERK2-mediated transcriptional activation requires ELK1 binding to the *EGR1* promoter. a** ELK1 consensus sequence (left), a potential ELK1 binding site in the *EGR1* promoter, boxed in red (middle), and the mutation introduced to generate the ELK1mut template in this study (right). **b** Immobilized template assay results showing decreased levels of ELK1, TOP2B, and CDK9 proteins on the ELK1mut template. INPUT, 25% of HeLa NE used for a reaction; DNA, the WT and ELK1mut immobilized template used for the assays. **c** Statistical representation of immobilized template assay results (*n* = 3 independent experiments). Data are presented as

mean values and SD. *P* values for the bar graphs were calculated with the unpaired, one sided Student's *t* test. **d** Immobilized template assay results showing the effects of WT and mutant ERKs on MED23 and Pol II recruitment to the WT and ELK1mut template (*n* = 2 independent experiments). **e** in vitro transcription assay results showing *EGR1* gene suppression on the ELK1mut template (*n* = 3 independent experiments). Data are presented as mean values and SD. *P* values for the bar graphs were calculated with the unpaired, one sided Student's *t* test. Source data are provided as a Source Data file.

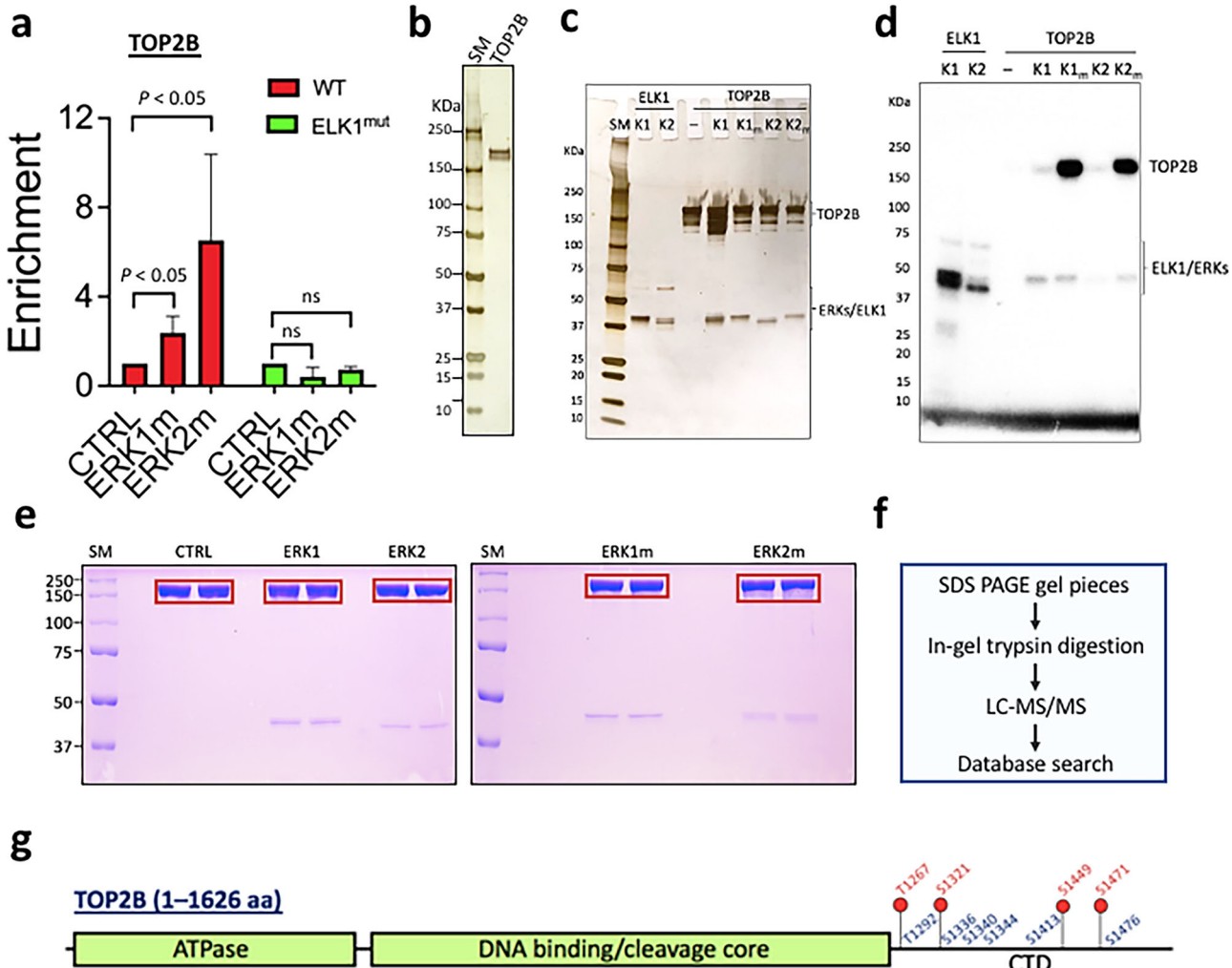

**Fig. 3 | TOP2B is phosphorylated by ERK1 and ERK2 proteins. a** Immobilized template assay results presenting TOP2B enrichment by ERK1m and ERK2 on the WT template but not on the ELK1$^{mut}$ template ($n = 4$ independent experiments). Data are presented as mean values and SD. $P$ values for the bar graphs were calculated with the unpaired, one sided Student's $t$ test. **b** Deubiquitinated TOP2B used for the in vitro kinase assays, silver-stained. **c** Silver-stained reactions following the in vitro kinase assay. **d** Autoradiograms of the in vitro kinase assay with TOP2B and ERKs. ELK1, a positive control. **e** In vitro kinase assay followed by SDS-PAGE for the preparation of mass spectrometry analyses. Red boxes showing sliced gel pieces used for the analyses. CTRL, ERK storage buffer only; SM, size marker (KDa). **f** A flow chart showing the steps of mass spectrometry analyses performed in this study. **g** A schematic representation of the TOP2B residues phosphorylated by ERK2/ERK2m (red), or both ERK1/ERK1m and ERK2/ERK2m kinases (blue). Source data are provided as a Source Data file.

human HEK cells and WT and mutant ERKs proteins were performed. Purified TOP2B and ERKs proteins used in the assays were silver-stained and are shown in Figs. 1b, d and 3b, c. Recombinant ELK1 (47 KDa) purified from *E. coli* was used as a positive control for ERKs (Fig. 3c; Supplementary Fig. 2A). After the kinase assay, a portion of each reaction was visualized by SDS-PAGE, followed by silver-staining (Fig. 3c), and analyzed by autoradiography (Fig. 3d; Supplementary Fig. 2B). As expected, the control, ELK1 was phosphorylated by both ERK1 and ERK2 (Fig. 3d). Although the intensity was milder, both ERK1 and ERK2 phosphorylated TOP2B (Fig. 3d). In contrast, TOP2B was intensively phosphorylated by the constitutively active ERKs, ERK1m and ERK2m (Fig. 3d; Supplementary Fig. 2B).

It seemed paradoxical TOP2B was phosphorylated by both ERK1 and ERK2 (Fig. 3d) although ERK1 and ERK2 showed differential effects on *EGR1* transcription (Fig. 1c–g). Thus, it was essential to understand if there are any differences in ERK1- vs ERK2-mediated TOP2B phosphorylation that allows these proteins to affect TOP2B differentially. To address this question, TOP2B was phosphorylated by ERK1, ERK1m, ERK2, and ERK2m in vitro and was sliced out from the gel after separation by SDS-PAGE (Fig. 3e). A negative control, HEK cell-purified,

in vitro dephosphorylated TOP2B itself was included (CTRL, Fig. 3e). The trypsin-digested TOP2B preparations were analyzed using mass spectrometry (Fig. 3e, f). In the assay, coverages for TOP2B peptides ($p < 0.05$) in the control and phosphorylated samples were 60–70% (Supplementary Data 1–5). We observed that some residues were pre-phosphorylated in the control TOP2B, and these residues were excluded from data collection. Our results indicated that ERK1 and ERK2 phosphorylated both mutual and distinctive residues on TOP2B (Fig. 3g; Supplementary Data 6). Notably, most of ERK-mediated phospho-sites were mapped to the CTD of TOP2B, including four tentative ERK2-specific phospho-sites (T1267, S1321, S1449, S1471) (Fig. 3g).

**Cryo-EM structure of the TOP2B-*EGR1* TSS-etoposide complex**
To investigate the mechanism by which TOP2B interacts with the *EGR1* DNA, we next performed cryo-EM analysis of the complex. In our previous study, multiple DNA segments of *EGR1* TSS (−423 to +332) were compared for binding to TOP2B, and the *EGR1* fragment ranging from −132 to +62 showed the highest affinity to TOP2B (Supplementary Fig. 3A)[47]. For the structural analysis, this 194 bp *EGR1* TSS was further

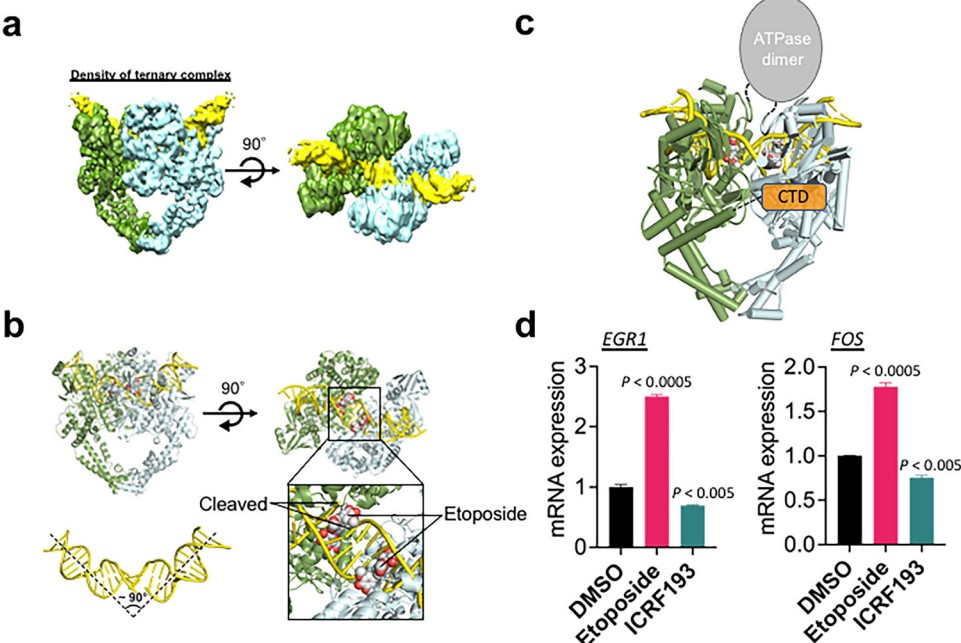

**Fig. 4 | Cryo-EM structure of the ternary complex of TOP2B-etoposide-*EGR1* TSS. a** Density maps of the ternary complex in three different angles. Two TOP2B proteins shown in green and cyan, *EGR1* TSS in yellow. **b** The ternary structure in ribbons. The DNA cleavage and etoposide binding sites, marked in a black box, are shown in close-up views. Two etoposide molecules are shown in sphere models, with the carbon and oxygen atoms colored white and red, respectively. The approximate angle of DNA bending is shown. **c** Schematic

representation of the entire TOP2B protein. **d** qRT-PCR results indicating transcriptional activation and repression at the *EGR1* and *FOS* genes in the HEK293 cells treated with etoposide and ICRF193, respectively (*n* = 3 biologically independent samples). Data are presented as mean values and SD. *P* values for the bar graphs were calculated with the unpaired, one sided Student's *t* test. Source data are provided as a Source Data file.

dissected into 4 segments to identify the DNA region spanning 40–50 bp with the highest affinity with TOP2B (Supplementary Fig. 3B). The resultant *EGR1* fragment (EGR1 #3-3, 50 bp, −30 to +20 of the TSS, Supplementary Fig. 3B) was used to form TOP2B-*EGR1* TSS complex. Although the protein-DNA complex itself was unstable, it could be stabilized by the addition of etoposide, a small chemical inhibitor that traps TOP2 in the TOP2-DNA cleavage complex, which is otherwise a transient intermediate.

The cryo-EM structure of the ternary complex was determined at a nominal resolution of 3.9 Å (Fig. 4a; Supplementary Fig. 4A; Supplementary Tables 2, 3). The structure contained a pseudo-symmetrical dimer of two TOP2B molecules (Fig. 4a–c; Supplementary Fig. 4E). Although the density of the ATPase domain of TOP2B (1–456 aa) was observed in the 2D and 3D reconstructions (Supplementary Fig. 4B, D), interpretable maps were not obtained in the 3D refinement probably because of its intrinsic flexibility relative to the core domain. Consequently, the flexible CTD (1208–1626 aa) containing ERK-phosphorylated residues was not visible. Therefore, the final model of TOP2B includes the DNA-binding and catalytic core domains (regions of 457–1118 aa and 1140–1207 aa, respectively). We observed a DNA segment docked to the central cleft of the TOP2B dimer (Fig. 4a–c; Supplementary Fig. 5A–C). Both of the two DNA strands were cleaved at the active site of each TOP2B, and the DNA is bent by approximately 90° (Fig. 4b; Supplementary Fig. 5A–C). Two etoposide molecules were intercalated into the two cleavage sites of the double strand of the *EGR1* TSS. These views are comparable to the previous crystal structures of the central part of TOP2B (residues 450–1206)[53]. The current complex structure shows a longer DNA stretch than those in the previous TOP2B structures, and two Lysine residues (K970 and K1011) potentially contact the DNA phosphate backbone (Supplementary Fig. 6). These interactions are conserved in TOP2A (Supplementary Fig. 6). Thus, the existence of CTD may barely affect the core structure of TOP2B and its DNA-binding mode. The CTD

phosphorylation may rather contribute, for example, to the TOP2B catalysis and the recruitment/dissociation to/from DNA[65,66].

When HEK293 cells were treated with 10 µM etoposide for 3 h, *EGR1* transcription was significantly increased (Fig. 4d). However, ICRF193, a small catalytic inhibitor of TOP2, decreased the transcription (Fig. 4d), as previously shown[4]. These results, together with the cryo-EM structure, strongly suggest that the DNA cleavage caused by TOP2B positively regulates transcriptional activation of the *EGR1* gene.

### ERKs regulate TOP2B catalysis to relax supercoiled DNA in distinctive manners

Next, the effects of ERK1 and ERK2 proteins on TOP2B were analyzed by DNA relaxation assays. The positively supercoiled pBR322 (4361 bp) was incubated with 0, 2, 40, and 100 nM TOP2B in the presence of ERK buffer only control (CTRL), ERK1, ERK1m, ERK2, and ERK2m at 200 nM at 30 °C for 6 min and then was separated on 1% agarose gel (Fig. 5a). The proteins after the relaxation reaction were shown by immunoblotting (Fig. 5b; Supplementary Fig. 7A). Compared with the control, more positively supercoiled (+SC) DNA was relaxed by TOP2B at 2 and 40 nM with both ERKs and their mutants (Fig. 5a). ERK2 and ERK2m appeared to enhance TOP2B catalytic activity for positively supercoiled DNA slightly more than ERK1 and ERK1m (Fig. 5a). Because transcription generates both positive and negative DNA supercoiling ahead and behind of Pol II[40,41], respectively, the effects of ERKs on TOP2B catalysis to resolve negative supercoiling were also investigated. The negatively supercoiled (−SC) pBR322 was incubated with 0, 1, 10, and 200 nM TOP2B along with CTRL, ERK1m or ERK2m for 6 min before separated on the gel (Fig. 5c). The results showed that ERK2m decreased the relaxation rate of −SC DNA by TOP2B at 1 nM, while ERK1m did not (Fig. 5c). These data suggested that ERK1 and ERK2 distinctively regulate the enzymatic activity of TOP2B to relax positive and negative DNA supercoiling, in particular, the negatively supercoiled DNA catalysis.

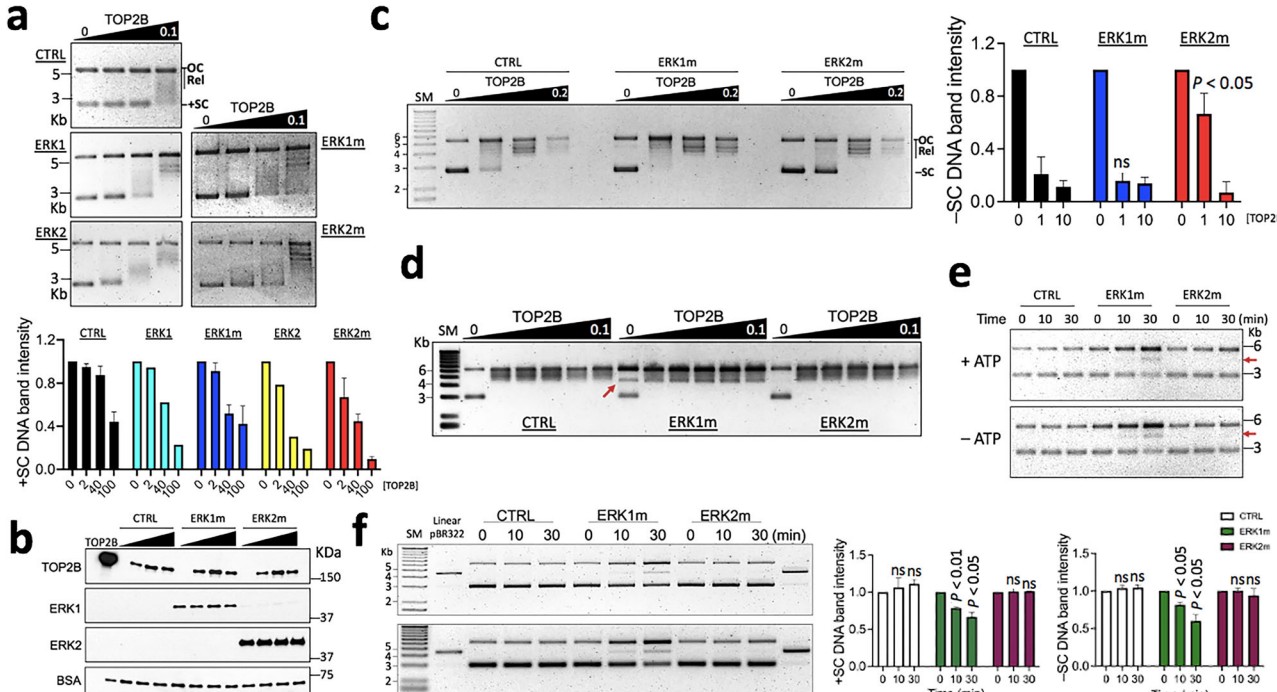

**Fig. 5 | ERK1 and ERK2 regulate TOP2B catalysis of DNA supercoiling in different manners. a** DNA relaxation assay results using positively supercoiled pBR322. The enzymatic reaction was allowed for 6 min before termination. TOP, representative agarose gel images showing the relaxation assay results. TOP2B concentrations shown in μM. OC, open circular DNA, Rel, relaxed DNA; +SC, positively supercoiled DNA. Bottom, a graph summarizing the relaxation assays ($n = 3$ independent experiments for CTRL, ERK1m, and ERK2m). TOP2B concentrations in nM. **b** Representative immunoblotting results showing the proteins used in the DNA relaxation assays. One tenth of a reaction was analyzed. TOP2B, 0, 2, 40, 100 nM; ERK1m and ERK2m, 20 nM. The loading control was TOP2B used for the reaction (TOP2B), 0.17 μM. **c** DNA relaxation assay results with negatively supercoiled pBR322. Left, a representative agarose gel image comparing CTRL, ERK1m, and ERK2m. −SC, negative supercoiled DNA. SM, DNA size markers in kilo base pairs (Kb). TOP2B concentrations in μM. Right, a graph summarizing the relaxation assay data ($n = 3$ independent experiments). TOP2B concentrations in

nM. **d** DNA relaxation assay results. The reaction time was allowed for 30 min. CTRL and ERK2m reactions showing a similar relaxation pattern, whereas ERK1m reactions showing a unique middle-sized DNA band (marked with a red arrow) with or without TOP2B. **e** DNA relaxation assay results suggesting supercoiled DNA relaxation by ERK1m to accumulate OC/relaxed and linearized/partially-relaxed plasmid DNA (red arrows) with or without ATP. Time, reaction duration. **f** DNA relaxation assay with positively and negatively supercoiled pBR322. Left, representative agarose gels showing the assay results. ERK1m decreases both positive (upper gel) and negative (bottom gel) supercoiling in the time course. The middle band formed by ERK1m might be the linearized pBR322 as they run similarly. Right, graphs summarizing the relaxation assay data ($n = 3$ independent experiments). Data in (**a**), (**c**), (**f**) are presented as mean values and SD. *P* values for the bar graphs in (**c**) and (**f**) were calculated with the unpaired, one sided Student's *t* test. Source data are provided as a Source Data file.

In addition, TOP2B was titrated at 0, 2, 4, 8, 40, and 100 nM with CTRL, ERK1m, and ERK2m for 30 min at 30 °C. In this longer reaction time, most +SC DNA was relaxed by TOP2B with or without ERKs. On the other hand, we noticed a band (marked with a red arrow) between the supercoiled and open circular (OC)/relaxed DNA bands in the presence of ERK1m without TOP2B (Fig. 5d). The unique sized, linearized or semi-relaxed form of DNA existed persistently in all ERK1m reactions, even with the highest TOP2B amount (Fig. 5d). Because the middle DNA band was shown in ERK1m reactions even without TOP2B, we tested ERK1m and ERK2m for their capability to relax the +SC DNA. The results showed that ERK1m can relax the +SC DNA by itself and it gradually increases the OC/relaxed and semi-relaxed/linearized DNA in 10 and 30 min reactions, with or without ATP (Fig. 5e). The reactions were repeated with both +SC and −SC DNA, confirming that ERK1m converts a portion of the supercoiled plasmid DNA into the OC/relaxed one (Fig. 5f; Supplementary Fig. 7B). The linearized pBR322 included in the gel electrophoresis suggested that the unique middle band formed by ERK1m might be a linearized plasmid although its identity is still not completely clear (Fig. 5f; Supplementary Fig. 7B). For these unexpected topological effects by ERK1m, we suspected any potential contaminations of bacterial nucleases, and examined the CTRL, ERK1m, ERK2m protein reactions and preps by screening mass spectrometry data and silver staining of PAGE gels (Fig. 1d; Supplementary Data 7–9). These examinations rarely showed supporting evidence of

nuclease contaminations, specific to ERK1m. Overall, our in vitro data suggest that activated ERK1 might possess the ability to relax supercoiled DNA on its own.

## ERK2-mediated phosphorylation of TOP2B regulates TOP2B release at the *EGR1* gene

We sought to understand the function of ERK2-mediated phosphorylation of TOP2B during transcriptional activation in the cell via inhibition of the catalytic activity of ERK2. VX-11e (VX11) is a small chemical inhibitor, specific to ERK2[67,68]. HEK293 cells were treated with 100 nM VX11 for 3 h. The *EGR1* mRNA expression was measured by reverse transcription followed by real-time quantitative PCR (qRT-PCR). The results indicated that VX11 effectively reduced *EGR1* transcription (Fig. 6a). *HSP70*, a control gene that is not an ERK2 target gene, was not affected by these conditions (Fig. 6a). The inhibitory effect of VX11 was more dramatic when *EGR1* was transcriptionally activated by serum induction for 15 min (DMSO/S15 vs. VX11/S15) after synchronizing the cell cycle at G0 by serum starvation (DMSO/S0, VX11/S0)(Fig. 6b). Similar results were shown for the transcription of the *FOS* gene, another IEG and ERK2-target gene, that VX11 treatment interfered with *FOS* mRNA synthesis (Fig. 6b). Moreover, S2 Pol II occupancy at *EGR1* was notably reduced in the VX11/S15 samples, compared with the DMSO/S15 control (Fig. 6c). TOP2B in the *EGR1* TSS became increased upon serum induction (DMSO/S0 vs DMSO/S15, Fig. 6d), as it was

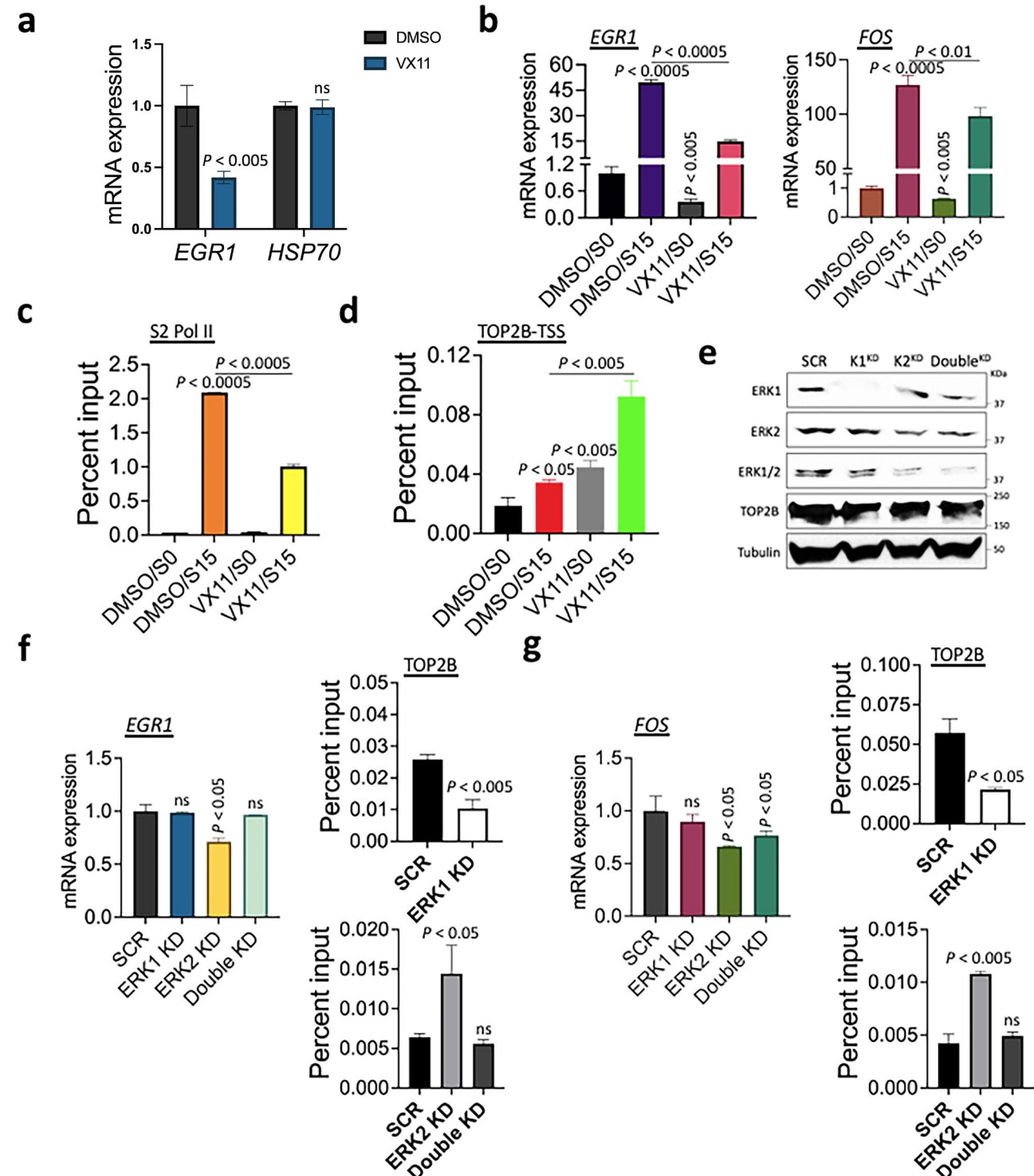

previously demonstrated that TOP2B is recruited upon transcriptional activation in IEGs. Strikingly, VX11 treatment markedly increased TOP2B occupancy even in the transcriptional resting state (VX11/S0), to a degree much higher than that of the DMSO control. Serum induction in the presence of VX11 further increased *EGR1* TSS-bound TOP2B (Fig. 6d).

Since we hypothesized that transcriptional repression by VX11 might counteract the recruitment of TOP2B in transcriptional activation, the increased TOP2B occupancies with VX11 treatment were surprising and required further validation. Thus, we downregulated ERK2 and quantitatively measured the occupancy of TOP2B associated

with the *EGR1* TSS. ERK KD using small interfering RNA species targeting ERK1 (K1[KD]), ERK2 (K2[KD]), or both proteins (Double[KD]) was performed in HEK293 cells. Immunoblotting using anti-ERK1, -ERK2, and -ERK1/2 antibodies confirmed a reduction in targeted proteins (Fig. 6e). qRT-PCR and ChIP-qPCR data comparing these cells indicated that ERK2 KD decreased *EGR1* mRNA production and dramatically increased TOP2B occupancy with the *EGR1* TSS, respectively (Fig. 6f). On the other hand, ERK1 or double KD did not affect the *EGR1* expression while ERK1 KD decreased TOP2B occupancy and double KD did not affect TOP2B occupancy in the *EGR1* TSS (Fig. 6f). Similar results were obtained for the *FOS* gene: ERK2 KD decreased the

**Fig. 6 | ERK2 function is required for normal TOP2B behavior and transcriptional activation. a** qRT-PCR results presenting the effects of ERK2 catalytic inhibition on *EGR1* and *HSP70* transcription. HEK293 cells were not synchronized (*n* = 3). Data are presented as mean values and SD. *P* values for the bar graphs were calculated with the unpaired, one sided Student's *t* test. **b** qRT-PCR results presenting the effects of ERK2 catalytic inhibition on the *EGR1* and *FOS* transcription (*n* = 3). HEK293 cells were synchronized at $G_0$ (S0) before they were serum-induced to early $G_1$ (S15). Data are presented as mean values and SD. *P* values for the bar graphs were calculated with the unpaired, one sided Student's *t* test. **c** ChIP-qPCR results showing S2 Pol II occupancy changes in the *EGR1* gene with or without functional ERK2 (*n* = 3 independent experiments). Data are presented as mean values and SD. *P* values for the bar graphs were calculated with the unpaired, one sided Student's *t* test. **d** ChIP-qPCR results showing a dramatic increase of TOP2B at

*EGR1* upon the catalytic inhibition of ERK2 (*n* = 3 independent experiments). Data are presented as mean values and SD. *P* values for the bar graphs were calculated with the unpaired, one sided Student's *t* test. **e** Immunoblots of ERK1 (K1KD), ERK2 (K2KD), and ERK1/ERK2 double KD (DoubleKD) HEK293 cells. SCR, scrambled siRNA; Tubulin, a reference. **f** qRT-PCR and ChIP-qPCR results presenting the effects of ERK KDs on *EGR1* transcription (left) and TOP2B occupancies at the *EGR1* TSS (right), respectively (*n* = 3 independent experiments). Data are presented as mean values and SD. *P* values for the bar graphs were calculated with the unpaired, one sided Student's *t* test. **g** qRT-PCR and ChIP-qPCR results presenting the effects of ERK KDs on *FOS* transcription (left) and TOP2B occupancies at the *FOS* TSS (right), respectively (*n* = independent experiments). Data are presented as mean values and SD. *P* values for the bar graphs were calculated with the unpaired, one sided Student's *t* test. Source data are provided as a Source Data file.

**Fig. 7 | A model of TOP2B regulation in IEG transcription.** During Pol II pausing at *EGR1*, ELK1 and ERK2 are inactive. TOP2B is ubiquitinated by the BRCA1/BARD1 complex (purple/pink circles)[47]. Upon transcriptional activation, ERK2 is activated to phosphorylate ELK1 and TOP2B (phosphorylation marked with a black star and a letter "P"). TOP2B generates DNA bending, relaxes DNA supercoiling, mediates DNA strand break (DSB), triggering DNA damage response signaling that phosphorylates and activates multiple enzymes (tan circles) including PI3K, TRIM28, and BRCA1[4,21,47]. If ERK2 is catalytically null or absent, gene activation is hindered and TOP2B is abnormally increased in the *EGR1* TSS. Nascent RNA, an orange curved line.

gene expression and increased TOP2B association (Fig. 6g). Collectively, the results from ERK2 catalytic inhibition by VX11 and KD experiments suggest that the absence of functional ERK2 results in abnormal TOP2B accumulation, repressing transcription at *EGR1* gene.

## Discussion

Our previous and current studies have characterized a role for TOP2B and associated protein factors including the BRCA1-BARD1 complex, ELK1, and ERKs to regulate TOP2B in IEG transcription[4,5,47] (Fig. 7). In Pol II pausing at *EGR1* before transcriptional activation, the BRCA1-BARD1 complex functions as an E3 ligase to ubiquitinate and stabilize TOP2B on the DNA[47]. While ELK1 is bound on the *EGR1* promoter and ERK2 exists in the nucleus (Figs. 1e and 2b, c), they are not yet activated[69,70]. Upon transcriptional activation, the MAPK pathway is triggered to phosphorylate and activate ERK2[56]. ERK2, in turn, phosphorylates ELK1 and TOP2B and modulates TOP2B activity to relax DNA supercoiling and TOP2B dissociation from DNA (Figs. 3 and 5–7). TOP2B catalyzes the DNA, which produces DNA break (Fig. 4) and provokes the activation of phosphoinositide 3 (PI3) kinases-related kinases[4,10,47]. ATM and DNA-PK phosphorylate different substrates

including BRCA1, DNA-PK, TRIM28, and γH2AX[21,47] (Fig. 7). Although the function of such phosphorylation is incompletely understood, our data suggested that phosphorylated BRCA1 at S1524 destabilizes TOP2B[47] and inhibiting ERK2 kinase activity results in accumulation of TOP2B (Fig. 6), suggesting that the post-translational modification status of TOP2B (e.g. phosphorylation and ubiquitination) modulates the enzyme (TOP2B)-substrate (DNA) affinity and/or catalytic cycle.

ERK1 and ERK2 are considered to have redundant cellular roles[58,61], although some studies have reported their different, even opposite functions in certain biological pathways, such as cell proliferation[57,59,60,71,72]. Some studies have investigated the foundation of these differences between ERK1 and ERK2 proteins. For example, ERK1 and ERK2 have dramatically different trafficking or shuffling rates from the cytosol to the nucleus, which is due to the less conserved N-terminal domains between these two proteins[73]. In another previous study, an N-terminal deletion mutant (Δ19–25) functioned as a dominant negative mutant for the endogenous ERK2[74], suggesting the importance of the N-terminal region for the unique nature of ERK2. In this study, we examined ERK1 and ERK2 and presented their distinct roles in IEG transcriptional activation. The results suggested that

ERK2 activates *EGR1* transcription, whereas ERK1 suppresses it (Figs. 1c, f, g, 6f; Supplementary Fig. 1), which provides the mechanistic explanation for previous observations of the ERK2 role in cell proliferation[57,59,71]. It is unclear whether the N-terminal domains of ERK1 and ERK2 contribute to their distinctive roles in the transcriptional activation of IEGs. In this context, it will be important to understand the differential and mutual functions, interactomes, and dynamics of ERK1 and ERK2 in the future.

In our biochemical analyses, ERK2 enhances the catalytic activity of TOP2B to relax positive DNA supercoiling (Fig. 5a, b). For negative DNA supercoiling, ERK2 could retard the relaxation mediated by TOP2B at a low concentration (Fig. 5c). During transcriptional activation, negative supercoiling is reportedly amplified in the promoter, which potentiates gene activity[75,76]. This suggests that ERK2 may function as a transcriptional activator by slowing-down TOP2B catalytic activity to relax the negative supercoiling in the promoter, while it enhances TOP2B catalysis of positive supercoiling, presumably formed ahead of paused Pol II[41,77], upon transcriptional activation. It is noted that excessive supercoiling, both positive and negative, is unfavorable for Pol II translocation[41], suggesting that the degree of negative supercoiling in the promoter during transcriptional activation may be controlled, rather than unleashed, for effective transcription. In addition, we found that the constitutively active ERK1 mutant is capable of relaxing the positive and negative supercoiling of DNA and accumulates relaxed/nicked/linearized DNA by itself independent of ATP (Fig. 5d–f). ERK1 increases TOP2B catalysis of positive DNA supercoiling, while it doesn't affect that of negative DNA supercoiling (Fig. 5a–c). These data suggest the different, intrinsic effects of ERK1 and ERK2 on DNA topology and their distinctive interactions with TOP2B to modulate the catalytic activity of the enzyme. This finding may explain the activating and inhibitory function of ERK2 and ERK1, respectively, on transcriptional activation at the representative Pol II pausing-harboring IEGs, *EGR1* and *FOS* genes (Figs. 1 and 5–7). Further studies will be important to reveal the dynamics and detailed mechanisms of differential interactions of ERK1 and ERK2 with TOP2B in Pol II pausing regulation.

ERK1 and ERK2 are the key kinases in the MAPK pathway that promote cell growth. ELK1 and MYC are among the targets of ERK1 and ERK2[31,56,64]. Other studies have suggested an interaction between ERK1/ERK2 and TOP2[62,78]. While a study showed that ERK2 phosphorylates and activates TOP2A, it also identified that a diphosphorylated catalytic mutant version of ERK2 still activates TOP2A[62], suggesting that the interaction between phosphorylated ERK2 and TOP2A, not TOP2A phosphorylation by ERK2, is required for TOP2A activation. In another study, TOP2 poisoning increased ERK1/2 phosphorylation and ERK1/2 inhibition reduces TOP2 poisoning-induced cell cycle arrest[78]. It should be noted that, unlike the earlier dogma that TOP2A is involved in replication whereas TOP2B is in transcription, a recent study suggested that the role of TOP2A is important for transcriptional regulation[36]. In addition, both ICRF193 and etoposide inhibit both TOP2A and TOP2B, imposing difficulty to distinguish their roles in cell-based analyses using such inhibitors. These considerations allow the possibility that ERKs might also interact with TOP2A for transcriptional regulation, an important hypothesis which awaits future studies.

In our current study, we have shown that TOP2B is a substrate of ERK1 and ERK2 (Fig. 3; Supplementary Data 1–6). Interestingly, a couple of sites are mutually phosphorylated by both the kinases while four residues are uniquely phosphorylated by ERK2 in the TOP2B CTD (Fig. 3g). It is noted that HEK-purified TOP2B was substantially phosphorylated, which interfered with screening the ERK-mediated phosphorylation sites by mass spectrometry, even after the effort to strip them off by phosphatases in vitro. Further mutational analyses are required to validate and characterize the multiple phospho-sites suggested by this study in future. Since ERK2 catalytic inhibition using VX11 caused a dramatic increase in TOP2B occupancies in the *EGR1* TSS

(Fig. 6d), the data suggest that the lack of ERK2-dependent phosphorylation of TOP2B at these sites may account for this unusual accumulation of TOP2B. Consistently, ERK2 KD, but not ERK1 or double KD, increased TOP2B in the TSSs of *EGR1* and *FOS* genes (Fig. 6f, g). It is unclear at this stage what is the nature of the accumulated TOP2B and whether the abnormal increase of TOP2B is direct and/or indirect consequence of missing functional ERK2. These are important questions to be answered in future. Perhaps, these are the TOP2B molecules that might be either abnormally recruited or trapped (unable to be released from) on the DNA. TOP2B recruitment and catalysis of DNA double-strand breaks typically occur in the TSS of IEGs, leading to IEG activation (Figs. 3a and 6b)[4,10]. However, in the case of ERK2 catalytic inhibition and KD, IEG expression was hindered (Fig. 6a–d, f, g), suggesting that TOP2B is not activated to catalyze the DNA but stays on the DNA in the absence of functional ERK2. This interpretation is supported by the observations that ICRF193, which causes TOP2B to be stuck on DNA without catalysis[79,80], inhibits transcription at *EGR1* and other IEGs (Fig. 4d)[4].

In contrast to ICRF193, etoposide treatment induced gene activation in *EGR*1 and *FOS* (Fig. 4d). In the structural data obtained, the *EGR1* TSS associated with the core TOP2B domain and etoposide exhibited a dramatic bending of DNA, approximately 90°, each DNA strand cleaved by a TOP2B protein (Fig. 4a–c; Supplementary Fig. 4,5). The bending and break of the TSS may bring the promoter and the TSS, where Pol II is paused in stimulus-inducible genes, to a proximity with increased flexibility (Fig. 7). This would allow DNA-binding transcriptional activators in the promoter to interact with the paused Pol II in the TSS, reverse pausing and activate transcription. DNA bending has been reported to play a positive role in transcriptional activation[81–84]; however, the exact roles of DNA break and DDR in transcription are not well understood. Although topological stresses during transcription require the function of TOP1 and TOP2[4,10,25,26,34,47], TOP2-mediated DNA catalysis does not typically result in DDR[25]. In spite of the dogma, transcription is coupled with DDR signaling and the activation of DNA repair enzymes, and the catalytic function of TOP2B contributes to transcription and DDR activation[4,10,21,25,47]. Thus, it is likely that DDR signaling is activated when TOP2B is removed from the broken site without rejoining it to expose the lesion. To the best of our knowledge, it is not clear whether DNA breaks and DDR are byproducts of transcriptional activation or play an active role in the process (chicken or the egg dilemma). However, it should be noted that inhibiting DDR or critical DNA repair enzymes interferes with Pol II pause release and gene expression[4,10,14,21,47,85], even within minutes in previous studies[4,21], suggesting a proactive role of DNA break and DDR in transcriptional activation. Moreover, it is not difficult to find examples of purposeful and functional DNA fragmentation and small nucleic acid production in DNA metabolic processes such as DNA replication[86], recombination[87], and repair[88]. In the future, it will be important to understand the role of TOP2-mediated DNA break and the mechanisms of TOP2 regulation for gene activation.

## Methods

### Cell culture and conditions
Cell culture and serum induction of HEK293 were carried out as previously described[4,47,89]. For TOP2B inhibitors, ICRF193 (Sigma, I4659) and etoposide (Sigma, 341205) were used at a final concentration of 10 μM in 0.1% DMSO for 3 h. An ERK2 inhibitor, VX-11e (Selleck, S7709) was used at a final concentration of 100 nM in 0.1% DMSO for 3–5 h. Control cells were treated with the same amount of DMSO for the same duration. Serum starvation and induction were performed as previously described[4,47].

### Cell transfection and knock-down experiment
HEK293 cells were grown to 60–70% confluence in a complete medium. The medium was replaced with Opti-MEM (Gibco, 31985) before

transfecting with scrambled siRNA (Santa Cruz Biotechnology, sc-37007) and ERK1 siRNA (Cell Signaling Technology, #6436), ERK2 siRNA (Santa Cruz Biotechnology, sc-35335), or both ERK siRNA species. Lipofectamine 2000 (Invitrogen) or FuGENE HD (Promega, E2311) transfection reagent was used according to the manufacturer's instructions. The transfected cells were collected after 48 h or 72 h incubation for RNA or protein/ChIP analyses, respectively. HEK293 cell extracts were prepared using a RIPA buffer (Cell Signaling Technology) in the presence of freshly added protease inhibitors [1 mM benzamidine, 0.25 mM PMSF, 1 mM Na-metabisulfite, 1 mM dithiothreitol (DTT); chemicals purchased from Sigma]. Protein concentrations in the cell extracts were determined by Bradford assays (Bio-Rad Laboratories) using BSA standard curves.

### Real-time PCR
Total RNA was purified using the RNeasy kit (Qiagen) following the manufacturer's instructions. The concentration of RNA was measured using Nanodrop and 0.6 μg of purified RNA was converted into cDNA by reverse transcription using ReverTra Ace qPCR RT Master Mix (Toyobo). Real-time PCR was conducted with equal amounts of resultant cDNAs, the indicated primers (Table S1), and SYBR Green PCR Master Mix (Applied Biosystems) using the StepOnePlus Real-Time PCR System (Applied Biosystems). β-Actin was used as the normalizer. Thermal cycles were as follows: 1 min at 95 °C followed by 45 cycles of 15 s at 95 °C, 15 s at 55 °C, and 1 min at 72 °C. The results were presented as relative fold differences, standard deviations (SDs), and statistical validations (see below).

### DNA template construction
The *EGR1* template DNA, −423 to +332, was cloned into a pCR-Blunt-TOPO to generate pTOPO-EGR1[47]. The biotinylated *EGR1* template was PCR-amplified using a pair of primers, one conjugated with biotin at the 5′ end (Table S1), and prepared using the same method described in our previous study[47]. The ELK1[mut] template was constructed using a set of primers (Table S1), introducing the desired mutation, and pTOPO-EGR1 as a template. The resultant ELK1[mut] construct was sequenced for validation. All primers used in this study were purchased from Integrated DNA Technology.

### HeLa NE and protein preparation
HeLa nuclei were provided by Dr. D. J. Taatjes at the University of Colorado Boulder. HeLa NE was prepared as previously described[21,47]. WT and mutant ERK1 (human) and ERK2 (mouse) bacterial expression vectors were generated from a previous study[63] and gifted by Drs. N. Soudah and D. Engelberg at the Hebrew University of Jerusalem. To express human ERK2, we incorporated two amino acids and a point mutation, which are the only differences between human and mouse ERK2 proteins, into the WT and mutant ERK2 plasmids using two pairs of primers (Table S1). BL21 cells were grown in Luria-Bertani (LB) medium (Qmbrothia) containing 50 mg/L kanamycin (Applichem). The expression of recombinant ERK protein was induced by 0.3 mM isopropyl-β-D-thiogalactopyranoside (IPTG) at an optical density of 0.6. Cells were further incubated at 20 °C for 18 h. Cells were harvested, resuspended in the binding buffer (50 mM Tris pH 8.0, 150 mM NaCl, 5 mM β-mercaptoethanol) and lysed by ultrasonication (SONICS, VCX-500/750), followed by cell debris removal by centrifugation. The supernatant was collected and loaded onto a Ni-NTA HiTrap chelating column (GE Healthcare). After eluting the bound proteins with an elution buffer containing 500 mM imidazole (50 mM Tris pH 8.0, 150 mM NaCl, 5 mM β-mercaptoethanol, 500 mM imidazole), the target protein was further purified using a HiPrep 16/60 Sephacryl S-300 HR column (GE Healthcare). The size-exclusion column buffer contained 50 mM Tris pH 7.5, 150 mM NaCl, 1 mM DTT. Human full-length TOP2B was purified from HEK cells, as previously described[47,90]. For the in vitro kinase assay and cryo-EM, ubiquitin-stripped TOP2B[47] was

used. The activity of purified TOP2B was confirmed using an in vitro decatenation assay, as previously described[47]. To obtain dephosphorylated TOP2B used for in vitro kinase assay and mass spectrometry the YFP column incubated twice for 15 min at room temperature in wash buffer supplemented with 0.1 mM MnCl$_2$ and 5 units mL$^{-1}$ calf intestinal phosphatase (NEB), 400 units mL$^{-1}$ lambda protein phosphatase (NEB), and 0.1 μg mL$^{-1}$ protein phosphatase 2a (Cayman Chemical) phosphatases, followed by 4 washes of 2 column volumes each to remove phosphatase enzymes. After elution from the anti-YFP affinity column, TOP2B proteins were further purified by sequential cation exchange chromatography using a 6 mL Source 15 S column (Cytiva) and size-exclusion chromatography using a HiPrep Superdex 200 16/60 column (Cytiva). Full-length ELK1 gene was amplified from pCGN-ELK1 purchased from Addgene (plasmid #27156) using a set of primers (Table S1) including XhoI and NdeI restriction sites and KOD plus NEO polymerase (Toyobo). The ELK1 PCR product was digested with the above restriction endonucleases (New England Biolabs) and ligated to the pET17b plasmid, which was linearized with the same enzymes, using T4 DNA ligase (New England Biolabs). The cloned ELK1 was validated by sequencing (Macrogen). *E. coli* BL21 strain was transformed with the resultant plasmid, pET17b-ELK1. The bacterial cells were grown to OD$_{600}$ 0.3–0.4 and ELK1 protein expression was induced at 30 °C for 5 h using 0.2 mM IPTG. The cells were lysed with the xTractor Bacterial Cell Lysis Buffer (Clontech) and sonicated. His$^6$-tagged ELK1 was purified on a Ni affinity column using the same method as previously described[21,47]. The purified proteins were validated by SDS-PAGE and immunoblotting (see the antibody information below).

### Immobilized template assay and Immunoblotting
The procedure and reagents for the immobilized template assay were identical to those used in our previous studies[21,47]. Dynabeads M-280 Streptavidin (Invitrogen) was washed and equilibrated with 2× B&W buffer (10 mM Tris-HCl, pH 7.5, 1 mM EDTA, 2 M NaCl) and incubated with the biotin-conjugated *EGR1* template DNA (−423 to +332) at 10 ng DNA/μL beads. The template-conjugated beads were washed twice with 1× B&W buffer and twice with 0.1 M Buffer D1 (20 mM HEPES, pH 7.6, 20% glycerol, 0.1 mM EDTA, 100 mM KCl). For each reaction, 120 ng of the immobilized template was resuspended in TF buffer (12.5 ng/μl dI-dC, 0.075% NP40, 5 mM MgCl$_2$, 250 ng/μL BSA, 12.5% glycerol, 100 mM KCl, 12.5 mM HEPES, pH 7.6, 62.5 μM EDTA, 10 μM ZnCl$_2$) for pre-incubation with WT or mutant ERK proteins for 30 min. The pellet was resuspended in NE buffer (17.5 ng/μL dI−dC, 0.1% NP40, 7.5 mM MgCl$_2$, 1.25 μg/μL BSA, 8.7% glycerol, 8.7 mM HEPES, pH 7.6, 44 μM EDTA, 130 mM KCl, 10 μM ZnCl$_2$). HeLa NE and purified recombinant WT or mutant ERK proteins at 100 ng/reaction, when indicated, were incubated with agitation for 40 min at room temperature (RT) to assemble the PIC. The template-protein complex was washed with a 10 beads volume of TW buffer (13 mM HEPES, pH 7.6, 13% glycerol, 60 mM KCl, 7 mM MgCl$_2$, 7 mM DTT, 100 μM EDTA, 0.0125% NP40, 10 μM ZnCl$_2$) and then the proteins were extracted using TW buffer including 2% sarkosyl or SDS-loading buffer for protein identification and quantification by SDS-PAGE and immunoblotting. For in vitro transcription assay, after TW wash, the beads were resuspended in TC buffer I (13 mM HEPES, pH 7.6, 13% glycerol, 60 mM KCl, 7 mM MgCl$_2$, 10 μM ZnCl$_2$, 7 mM DTT, 100 μM EDTA, 15 ng/μL dI-dC, 10 mM creatine phosphate). TF, NE, TW, and TC buffers were supplemented with freshly added protease, 1 mM benzamidine, 0.25 mM PMSF, 1 mM Na-metabisulfite, 1 mM DTT, and aprotinin (Sigma A6279, 1:1000). The primary antibodies used in this study were Pol II (ab817, Abcam; #2629, Cell Signaling Technology; A304-405A, Bethyl Laboratories), CDK9 (sc-13130, Santa Cruz Biotechnology), MED23 (A300-425A, Bethyl Laboratories), ELK1 (sc-365876, Santa Cruz Biotechnology; #91825, Cell Signaling Technology), ERK1/2 (ab17942, Abcam), ERK1 (sc-271269, Santa Cruz Biotechnology), ERK2 (ab32081,

Abcam) and TOP2B (A300-949A, Bethyl Laboratories; sc-25330, Santa Cruz Biotechnology).

## In vitro transcription and RNA quantification

For in vitro transcription assay, the procedure and reagents were identical as previously described[21,47], except for the method used to quantify the in vitro synthesized *EGR1* mRNA molecules. A mixture of NTP at a final concentration of 250 μM A/G/C/U was added to the reactions prepared using the immobilized template assay. The mixture was incubated at 30 °C for 30 min to allow Pol II to polymerize mRNA molecules. When ERK proteins were assayed, the purified proteins were included during the PIC assembly and after 3–5 min of NTP addition. The polymerization reaction was proceeded for 30 min. Subsequently, 1.5 Kunitz unit DNase I (Qiagen) was added to the reaction and allowed to sit for an additional 15 min to digest the template DNA. The reaction was terminated with five volumes of 1.2× Stop buffer (0.6 M Tris-HCl, pH 8.0, 12 mM EDTA, 100 μg/mL tRNA). The pellet fraction, including the magnetic beads and the supernatant, was separated using a magnetic stand (Invitrogen). The supernatant was treated with an equal volume of phenol: chloroform: isoamyl alcohol (25:24:1, Sigma) solution to extract proteins. The soluble fraction, including RNA, was precipitated with 2.6 volumes of 100% ethanol (Sigma). Following centrifugation at 18,000 × g for 30 min, the pellet was dissolved in nuclease-free water. The RNA was incubated at 70 °C for 10 min, followed by centrifugation at 10,000 × g for 1 min. The cDNA construction mixture included 5 mM MgCl₂, 10× buffer, 1 mM dNTP mixture, AMV reverse transcriptase, random primers, and RNA, according to the manufacturer's instructions (Promega). The PCR conditions were 25 °C for 10 min, 42 °C for 60 min, and 95 °C for 5 min. The resulting cDNA was quantified using SYBR Green (Applied Biosystems) and real-time quantitative PCR (Applied Biosystems), using a primer set for probing *EGR1* mRNA (Table S1).

## Electrophoretic mobility shift assay (EMSA)

*EGR1* TSS double-stranded fragments, EGR1 #3-1, #3-2, #3-3, and #3-4 and *EGR1* TSS ssDNA were purchased from Integrated DNA Technology (Tables S1). The TOP2B dilution, DNA-protein binding, and gel electrophoresis/silver staining were performed as previously described without modifications[47].

## Cryo-EM sample preparation and image processing

*EGR1* TSS double-stranded DNA was produced by annealing of two complementary DNA strands: 5′-GAGTCGCGCGAGAGATCCCAGCGCGC AGAACTTGGGGAGCCGCCGCCGCCAT-3′ and 5′-ATGGCGGCGGCGG CTCCCCAAGTTCTGCGCGCTGGGATCTCTCGCGACTC-3′ (Integrated DNA Technology, Table S1). For the TOP2B-DNA-etoposide complex formation, 8.3 μM TOP2B, 16.7 μM dsDNA, and 1.67 μM etoposide were mixed in a buffer containing 20 mM HEPES pH 8.0, 5 mM KCl, 5 mM MgCl₂, and 0.5 mM ATP. Quantifoil grids (R1.2/1.3, Cu, 300 mesh; Quantifoil Micro Tools) were glow-discharged for 1.5 min using a PIB-10 ION Bombarder (Vacuum Device Inc.) immediately before use. Two microliters of sample were applied to the glow-discharged grids, which were blotted for 3 s and plunge-frozen in liquid ethane using a Vitrobot Mark IV (FEI) operated at 4 °C and 100% humidity (blotforce 5; blottime 3).

## Cryo-EM data collection and image processing

Micrographs were acquired on a Krios G4 transmission electron microscope (ThermoFisher Scientific) with a K3 detector and a Bio-Quantum energy filter (Gatan) operated with a slit width of 15 eV. Data collection was automated using EPU software and micrographs were taken at a magnification of ×105,000 (0.83 Å per pixel) with a total dose of 49 e/A2, fractionated over 40 frames. A total of 2327 micrographs were acquired. Estimation of the contrast-transfer function was performed with Gctf[91]. Motion correction, 2D and 3D classification

steps, and further processing were performed using RELION 3.0[92]. Particle picking was done by using Warp[93] and Topaz[94], and those particles were merged while avoiding overlap. Particles were first extracted with 3x binning (2.49 Å/pixel), and junk particles were removed using a series of 2D and 3D classifications by using the RELION program. The first reference map for the 3D classification was reconstructed by using the RELION 3D initial model. The remaining particles were re-extracted with a pixel size of 1.245 Å/pixel, and then refined with masks around TOP2B. Bayesian polishing and CTF refinement were then applied. Masks were created using the UCSF Chimera[95]. During 3D refinement, C2-symmetry was imposed. We also performed 3D reconstruction without imposing the C2 symmetry (Supplementary Fig. 5C). There is no large difference between the densities with and without imposing the C2 symmetry.

## Model building and refinement

The refined map was used for model building. For the TOP2B-DNA-etoposide complex structure, we first docked the existing structural models into the density. We used the full-length TOP2B model deposited in the AlphaFold Protein Structure Database (Uniprot code: Q02880) as an initial TOP2B coordinates, and then removed disordered regions from the model. For the initial models of dsDNA and etoposide, we used the crystal structure of the TOP2B-DNA-etoposide complex, deposited in the Protein Data Bank (PDB code: 3QX3). As the nucleotide sequence could not be identified in the density map, the DNA model was tentatively built in that the middle of the DNA sequence was placed in the middle of the TOP2B DNA binding domain. The model was fitted into the refined map in UCSF Chimera[95] and manually edited in Coot[96], followed by real-space refinement with PHENIX[97], using secondary structures and base-pairing restrains. Weak density for the ATPase domain of TOP2B was observed (Supplementary Fig. 3B). Our 3D refinement using the mask focused on the ATPase domain did not yield high-quality map of this domain, probably because of its small size and flexibility. Therefore, we did not build a model of this domain. Thus, the final model contains amino-acid residues 445–1118 and 1140–1201 of TOP2B. The final models were validated in Molprobity[98] and the figures were generated using UCSF Chimera and PyMol (Schrödinger LLC).

## DNA relaxation assay

Positively and negatively supercoiled pBR322 was purchased from Inspiralis (Cat. POS5001 and S5001). The relaxation/in vitro kinase assay buffer included 50 mM Tris-HCl (pH 7.5), 125 mM NaCl, 10 mM MgCl₂, 5 mM DTT, 0.1 mg/mL BSA as final concentrations. ERK proteins, 200 ng per each reaction, or an equal volume of the ERK storage buffer (sizing column buffer described above, control) were used. Each reaction included 0.5 μg of positively or negatively supercoiled pBR322 as the substrate for TOP2B. To obtain the linearized plasmid, 0.25 μg of the plasmid was incubated with a restriction endonuclease HindIII (R0104S, New England BioLabs) at 37 °C for 3 h. Dephosphorylated, deubiquitinated TOP2B (0 – 0.2 μM) or an equal volume of the buffer control was supplemented as indicated. The catalytic reaction was induced by adding ATP to 1 mM final concentration and was allowed at 30 °C for indicated durations (0–30 min). The reaction was terminated by adding equal volumes of both cold STOP solution [40% glycerol, 100 mM Tris-HCl (pH 8.0), 10 mM EDTA, 0.5 mg/mL bromophenol blue] and cold PCI on ice. The aqueous, upper phase was loaded onto a 0.8–1% agarose gel. The DNA on the gel was visualized by staining in 0.01% ethidium bromide solution for 10 min and sufficient destaining in water. For immunoblotting, 10% of the reaction mixture was reserved and stored at −70 °C until the analysis.

## Chromatin immunoprecipitation and qPCR

ChIP-qPCR analysis was conducted following the previously described methods and reagents without modifications[4,21,47]. The antibodies used

for this study were phosphorylated S2 Pol II (ab5095, Abcam) and TOP2B (A300-949A, Bethyl Laboratories). The primer sets used are listed in Table S1.

## In vitro kinase assay

The substrate, 1.8 µg of TOP2B was incubated with approximately 0.2–0.5 µg WT or mutant ERK proteins as a kinase in a reaction. For the control reaction, TOP2B without any kinase (ERK storage buffer only) was prepared side-by-side. Kinase buffer included 25 mM Tris pH 8.0, 2 mM DTT, 500 µM cold ATP, 100 mM KCl, 10 mM $MgCl_2$, and 2.5 µCi γ-$P^{32}$ (PerkinElmer) labeled ATP to be final concentrations per a reaction. The reaction was incubated at RT for 1–3 h before it was terminated using an 8× SDS-loading buffer. One microliter of each reaction was run on a 7% SDS-PAGE gel and visualized by silver-staining (silver nitrate, Sigma). The rest of the reaction was subjected to SDS-PAGE followed by autoradiography. For mass spectrometry, only cold ATP without γ-$P^{32}$ labeled ATP was used and the reaction was allowed for 1 h at 30 °C and separated on a 7% SDS-PAGE gel. The bands corresponding to TOP2B were sliced for the further analysis.

## Mass spectrometry

The proteins in each gel slice were extracted by in-gel digestion. The gel slices were diced into 1 mm pieces, and subjected to reduction with 10 mM tris (2-carboxyethyl) phosphine hydrochloride (SIGMA), at 56 °C for 1 h, alkylation with 55 mM iodoacetamide at room temperature for 45 min in the dark, and then digestion with 500 ng of trypsin (Thermo Scientific) at 37 °C for 16 h. The resulting peptides were extracted with 1% trifluoroacetic acid and 50% acetonitrile. Mass spectra were obtained on an LTQ-Orbitrap Velos pro (Thermo Scientific) coupled to a nanoflow UHPLC system (ADVANCE UHPLC; AMR Inc.) with Advanced Captive Spray SOURCE (AMR Inc.). The peptides mixtures were fractionated by C18 reverse-phase chromatography (3 µm, ID 0.075 mm × 150 mm, CERI). The peptides were eluted at a flow rate of 300 nL/min with a linear gradient of 5–35% solvent B over 60 min. The compositions of solution A and solution B were 2% acetonitrile, 0.1% formic acid and 100% acetonitrile, respectively. The mass spectrometer was programmed to carry out 13 successive scans, with the first consisting of a full MS scan from 350 to 1600 m/z using Orbitrap (resolution = 60,000), the second to thirteenth consisting of data-dependent scans of the top ten abundant ions obtained in the first scan using ion trap with CID. Automatic MS/MS spectra were obtained from the highest peak in each scan by setting the relative collision energy to 35% CID and the exclusion time to 90 s for molecules in the same m/z value range. The raw files were searched against the Uniprot Human proteome database (2022 .05.25 downloaded) and cRAP contaminant proteins dataset using the MASCOT program (version 2.8; Matrix Science) via Proteome discoverer 2.5 (Thermo Fisher Scientific). The search was performed using carbamidomethylation of cysteine as a fixed modification, and oxidation of methionine, acetylation of protein N-termini, phosphorylation of serine threonine and tyrosine as variable modifications. The number of missed cleavages sites was set as 3.

## Statistics and reproducibility

Standard deviation was calculated and used to generate error bars. The Student's $t$ test was used to determine statistical significance ($P < 0.05$, one-sided). Graphs were generated using the Prism 8 software (GraphPad, Inc.). Representative experimental images of gels and autographs were repeated at least twice independently with similar results.

## Reporting summary

Further information on research design is available in the Nature Portfolio Reporting Summary linked to this article.

## Data availability

All data are available in the manuscript or as supplementary information. Source data are provided with this paper. The cryo-EM data generated in this study have been deposited in the Electron Microscopy Database (EMDB) and to the Protein Data Bank (PDB) under accession code EMD-34022 and PDB 7YQ8. The proteomics data generated in this study have been deposited in the jPOST and Proteome Xchange under accession code JPST002096 [https://repository.jpostdb.org/entry/JPST002096.0] and PXD040977 respectively. Source data are provided with this paper.

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

## Acknowledgements

We are grateful to D. Engelberg and N. Soudah at the Hebrew University of Jerusalem for their generosity in sharing the bacterial WT and mutant ERK1/2 expression vectors and to D.J. Taatjes at the University of Colorado for the gift of HeLa NE. We thank C.A. Austin at Newcastle University and S.K. Calderwood at Beth Israel Deaconess Medical Center/Harvard Medical School for their helpful discussions and edits. We also thank Y.H. Kim and his students at Kyungpook National University (KNU) for their assistance with handling radioactive materials and sharing equipment, and S. Lee, D. Kim, M. Seu, B. Kang, and the current members of the Bunch laboratory members at KNU for their technical assistance. H.B. thanks D.Y. Bunch, John, and J. Christ for their loving encouragement and support throughout the course of this work. This research was supported by grants from the Japan Society for the Promotion of Science (KAKENHI, JP20H05690) to S.S. and from the National Research Foundation (NRF) of the Republic of Korea (2022R1A21003569) to H.B.

## Author contributions

H.B. and D.K. performed the immobilized template and transcription assays. H.B. performed in vitro kinase assay and prepared mass spectrometry samples. D.K. carried out quantitative real-time PCR of ChIP and RNA analyses. M.N., H.E., and S.S. performed cryo-EM data collection and structure determination: the cryo-EM experiments were performed at the RIKEN Yokohama cryo-EM facility. R.N. performed mass spectrometry. A.C., M.S., H.V., and J.C. performed the protein purification. D.K. and J.J. performed cloning and mutagenesis. H.B. performed cell culture, drug treatment, K.D. experiments, ChIP, DNA relaxation assays, and immunoblotting. H.B. created the hypothesis, designed and coordinated the experiments, analyzed and curated the data, and wrote and revised the manuscript.

## Competing interests

The authors declare no competing interests.

## Additional information

**Heeyoun Bunch** [1,2] ✉, **Deukyeong Kim** [2,7], **Masahiro Naganuma** [3], **Reiko Nakagawa** [4], **Anh Cong** [5], **Jaehyeon Jeong** [1], **Haruhiko Ehara** [3], **Hongha Vu** [6], **Jeong Ho Chang** [6], **Matthew J. Schellenberg** [5] & **Shun-ichi Sekine** [3]

[1]Department of Applied Biosciences, Kyungpook National University, Daegu 41566, Republic of Korea. [2]School of Applied Biosciences, College of Agriculture & Life Sciences, Kyungpook National University, Daegu 41566, Republic of Korea. [3]Laboratory for Transcription Structural Biology, RIKEN Center for Biosystems Dynamics Research, 1-7-22 Suehiro-cho, Tsurumi-ku, Yokohama 230-0045, Japan. [4]RIKEN BDR Laboratory for Phyloinformatics, Hyogo 650-0047, Japan. [5]Department of Biochemistry and Molecular Biology, Mayo Clinic, Rochester, MN 55905, USA. [6]Department of Biology Education, Kyungpook National University, Daegu 41566, Republic of Korea. [7]Present address: Department of Biological Sciences, Korea Advanced Institute of Science and Technology (KAIST), Daejeon 34141, Republic of Korea. ✉e-mail: heeyounbunch@gmail.com

