## [Peer Review File · Nature Communications]

ERK2-topoisomerase II regulatory axis is important for gene activation in immediate early genesEditorial Note: Parts of this Peer Review File have been redacted as indicated to remove third-party material where no permission to publish could be obtained.

Reviewers' comments:

Reviewer #1 (Remarks to the Author):

The manuscript by Kim et al. describes the potential role of human topoisomerase II β isoform (TOP2B) and of the phosphorylation of its C-terminal domain by the MAP kinase ERK2 in the regulation of immediate early gene transcription (EGR1). Inhibition or knock-down of ERK2 indicate that ERK2 activates transcription by increasing TOP2B binding to the transcription start site of EGR1 and FOS1.

The phosphorylation analysis of TOP2B brings interesting information but some experiments need to be better explained and justified. Together with the cellular assays, which results are more convincing, this study would benefit from further biochemical analysis to clarify the role and regulation of TOP2B activities by the phosphorylation.

In addition to in vitro and cellular assays, the authors present a cryoEM reconstruction of the TOP2B structure (DNA binding/cleavage domain). However, the structural data as presented does not support the conclusions on the role of TOP2B. This part would require more in-depth analysis and higher resolution structures if achievable, to provide structural details on the specificities of the EGR1 TSS DNA template compared with previous structures. The structure of this domain alone is not sufficient to account for the multiple interactions occurring during the assembly of large transcription complexes.

Many of the data presented in this article would need a thorough overhaul and reorganization of the figures to clarify the conclusions of the study.

Specific comments and questions:

In vitro transcription experiments

- The authors are incubating the EGR1 DNA template with a HeLa nuclear extract to compare the levels of transcription activation upon addition of ERK1 and ERK2. What are the levels of ERK1 and ERK2 endogenous proteins for activated transcription compared with the in vitro assay? What are the levels of ERK1/2 in the HeLa nuclear extract relative to the amount of recombinant ERK1/2 added in the assay?
- Figure 1: Panel C displays histograms for the levels of proteins associated with DNA template assessed by immunoblotting (WB shown in supplementary figure 1). On the contrary, panel E shows the immunoblotting of the proteins. The figures (main and supplementary) should be reorganized. The conclusions drawn from these experiments do not seem completely supported by the data. In particular the levels of CDK9 or MED23 do not seem significantly higher than in the control experiments in panel E, or in supplementary 1 panel B.
- Could the authors explain how the value of enrichment is assessed for the "none" condition on panel 1C, to which other control or basal level is it compared?
- Panel 1F: the authors claim that ERK1 has an opposite effect than ERK2 but the EGR1 transcript level for ERK1 is within the same range as control level. It looks like ERK1 has no effect when compared to ERK2 and it seems difficult to conclude at this stage that they have an "opposite" effect based on this particular experiment.
- Figure 2/3: the organization of data in supplementary figure 1 relative to the main figure 2/3 is confusing. Supplementary figure 1B is cited numerous times and it seems like some panels could move in the main figures.
- The use of "m" for either a point mutation in the protein or a mutation in the DNA sequence is also confusing. The terms "CTRL", "NONE" and "WT" refer to different controls in each panel and should be more clearly explained in the figure legends.
- The authors claim that TOP2B association with EGR1 is positively correlated with ELK1 binding but this binding may be due also to the fact that Pol II has a deficient binding on ELK1m template, since TOP2B has been shown to associate with Pol II complexes.
- Supplementary figure 2A: could the authors explain the presence of 2 bands for the ELK1 recombinant sample? these 2 bands do not appear in figure 3D and the single band in figure 3D appears between the 37 and 50 kDa markers where there is no significant protein band in supplementary figure 2A.
- Supplementary figure 2B: the legend should explain that the image is a 32P autoradiography of a SDS-PAGE gel.
- Figure 3D: the intensity of TOP2B phosphorylation by WT ERKs is indeed very low, the authors should provide another image with only the WT ERKs to be able to compare with the control (no kinase)? What is the kinase/TOP2B ratio and would the signal be higher with an increased ratio?

The ERKs can indeed be used to identify more robustly the phosphorylation sites, but it seems important to measure the effect of the wt ERKs rather than of the ERKs to assess the effect of the kinases on the TOP2B.

Structural data processing

The data processing is insufficiently described.

- Particle picking: what was the rationale behind the use of 2 different particle picking softwares? How many particles were identified simultaneously by the 2 softwares? Could the authors explain how the particles were then extracted and pulled? How was the first 3D reconstruction obtained, in particular what is the ab-initio model if there is one?

- Initial and final models for the DNA binding/cleavage region:

The authors used the AlphaFold2 model for Top2B as an initial template for model building. Which regions were used to start the model building, provided that some domains are disordered and not properly modelled? (domain boundaries?)

What is the completeness of the model compared to the crystallographic structures of TOP2B DNA binding/cleavage domain? How does the final cryoEM model compare with the AlphaFold model in terms of completeness and RMSD?

- Although weaker, the density for the ATPase domain is visible in some of the 2D classes. The authors should consider running 3D classification without C2 symmetry to attempt to see this region. Running of a non-uniform refinement and/or homogeneous refinement in cryoSPARC with and without the C2 symmetry should be done to attempt to improve the resolution.

DNA: 3D classification without alignment with a mask focus on DNA region should be tried to improve density of DNA and see if the DNA sequence can be identified or if the density of DNA base pairs corresponds to an average density?

- Structural analysis:

- There is no figure showing the density at the site of cleavage supporting the claim that DNA is cleaved or that etoposide molecules are bound.

How many base pairs are visible in the density out of the 50 bp used as DNA template?

- As mentioned by the authors, several structures of the homologous human TOP2 have been solved, either individual domains using x-ray-crystallography or full-length proteins using cryoEM (ref 52, 53, 54). All structures show sharp bending of the DNA molecule and some also have doubly nicked DNA with etoposide molecules inserted at the cleavage sites.

The DNA bending and cleavage mechanism is an intrinsic mechanism of the Top2, regardless of the DNA sequence. Is the orientation and bending of the EGR1 TSS template differing from the published structures, and if so, how? Is there any particular geometry in this promoter region that would support the claim that this is how transcription activation is happening? Superimposition of structures and indication of the bending angles would be useful.

If no structural difference, is there any difference on the flexibility/molecular dynamics of this DNA sequence?

- The structural data do not seem to explain why treatment of HEK293 cells by etoposide shows increased EGR1 transcription, and why treatment by ICRF-193 (targeting the ATPase domain, not visible in the cryoEM density) shows decreased transcription. DNA bending and cleavage occur without etoposide.

- The authors could not reconstruct density for the C-terminal region that is bearing most of the phosphorylation sites, which makes sense since this region is disordered.

However, 2 sites common to ERK1 and 2 are located on the DNA binding/cleavage core that was reconstructed (Figure 3G). An overall 3.9 Å resolution is not high enough to distinguish phosphate groups but local resolution may be better on some regions. In the TOP2B protein produced in HEK cells, without in vitro phosphorylation, are these positions phosphorylated? Is it possible to distinguish phosphorylation on positions T715 or S728? On which sub-domains are these residues positioned?

Which positions are crucial for TOP2B recruitment on the EGR1 TSS? To probe this model, it would be interesting to analyze point mutations on phosphorylated positions specific to ERK2 in the CTD in the cellular assays, and their impact on the catalytic activities.

Reviewer #2 (Remarks to the Author):

In the present manuscript, Kim et al. report that ERK2, but not ERK1, is important for IEG transcriptional activation. They also identify a possible ELK1 binding site on the EGR1 gene that appears to be important for ERK2 transactivation at EGR1 and the enrichment of TOP2B on the EGR1 transcription start site (TSS). The CTD of TOP2B is noted to be extensively phosphorylated by both ERK1 and ERK2, while the failure of this phosphorylation of TOP2B is linked to the accumulation of TOP2B on TSS of IEGs.

The organization and findings of the manuscript are not always logically laid out and were difficult to follow at times. The primary conclusion of this work proposes that TOP2B association, catalysis, and dissociation on DNA are important for regulating transcription and that ERK2-mediated TOP2B phosphorylation may be important for this process. Although there are new findings that are interesting (such as TOP2B being a substrate for ERK1/2), many of the data are insufficient and at times inconsistent to support the stated claims.

Primary comments

1) There are number of issues concerning various aspects of the recruitment and transcriptional activation data. For example,

- In Fig. 1E, the constitutively active mutant ERK2m does not appear to be more proficient in recruiting Pol II, MED23 and CDK9 than WT ERK2; however, in Fig. 1F the ERK2m activated transcription more efficiently than ERK2 WT. This dichotomy should be explained.
- In Fig. S1, the effects of recruitment by the various ERK constructs seem inconsistent. For example, S1A shows a slight increase for CDK9 with ERK2, while Fig. 1E does not. By comparison, Fig. S1B shows no effect on MED23 or TOP2B recruitment by any of the ERKs. Also, why does ERK1 appear to antagonize MED23 recruitment on the mutant ELK1 sequence? Finally, why does ERK2 appear to stimulate the recruitment of MED23 in panel A, but have no effect in panel B? The inconsistencies between replicates are problematic.
- In Fig. 5E the intensity of ERK1/2 in K1KD and K2KD are significantly different. This raises a concern that ERK1 or ERK2 may not be knocked down efficiently. The authors should check this with a specific antibody for ERK1 and ERK2, respectively.
- In Fig. 2B, the signal for Pol II seemed to be stronger in the ELK1m group than WT. Is this assay performed without adding ERK proteins? If yes, then why in Supplementary Figure 1B is the signal for Pol II much weaker in the ELK1m group than in the WT? And in Fig. 2C, why should the decrease noted for TOP2B be more statistically significant than the decrease reported for ELK1 and CDK9, when the degree of the decrease is smaller for TOP2B than for the other two proteins?
- The data shown in Fig. 5F-G do not comport with the authors' claims. If the presence of ERK2 has a positive role on gene expression and a negative impact on TOP2B accumulation (for EGR1), then this should be seen with the double knockout (which looks like WT). These data also appear to run counter to those shown for Fig. 3A, where ERK2 is claimed to boost TOP2B recruitment.
- Figs. 1F, 2D, 2E, 3A, 4D, 5A-D, and 5F-G: the authors need to show examples of the raw data used to generate bar graphs, as well as more clearly point out which data are representative of each graph.

Overall, the inconsistencies in the data and/or their presentation make it difficult to reconcile many of the observed findings with the stated claims.

2) In Fig. S2B, the kinase data are worrisome. Kinases are typically robust enzymes. The authors need 0.5-1 μg of kinase to see a signal on 9 μg of substrate after a full hour of incubation. This result indicates that the specific activity of the kinases is quite low, which raises concerns about their use and specificity in other experiments.

3) In Fig. 3C, the band of TOP2B in the lane TOP2B/K1 looks quite different from the bands in

other lanes. Are there controls to ensure that the same amount of protein was loaded into all the wells?

4) Regarding the statement "However, in the case of ERK2 catalytic inhibition and KD, IEG expression was hindered (Figs. 5A–C, 5E–G), suggesting that TOP2B is not activated to catalyze the DNA but stays on the DNA in the absence of functional ERK2" (Discussion section): ERK2 has many substrates besides TOP2B, including Pol II and ELK1. Thus, VX11 treatment or knock down of ERK2 would change the phosphorylation state of various proteins, which may lead to a change in IEGs transcription and TOP2B occupancy at the TSS. Based on the data in Fig. 5 it is hard to tell whether TOP2B is indeed "not activated to catalyze the DNA but stays on the DNA in the absence of functional ERK2". The authors should compare the relaxation/cleavage activity of TOP2B with/without phosphorylation by ERK2 (and ERK1) in vitro. If TOP2B becomes more active following ERK2 treatment, then it might be more reasonable to make this claim.

5) The authors showed that both ERK1 and ERK2 could phosphorylate TOP2B, and most of their phospho-sites overlap. By contrast, ERK1 and ERK2 showed differential effects on EGR1 transcription. The authors should explain why the phosphorylation of residues 1342, 1403, 1473, and 1588 led to different effect of ERK2 from ERK1.

6) Only the 5' end of one strand of the DNA template used in the immobilized template assay was fixed by attaching to the beads, which means that the DNA template will always be relaxed without any supercoiling tension during transcription. The supercoiled state of DNA template may affect transcription, raising concerns that the enrichment of TOP2B and Pol II, CDK9, and MED23 in Fig. 1-3 may not be physiologically relevant. Please comment.

7) Regarding the statement "When HEK293 cells were treated with 10 μ M etoposide for 3 h, EGR1 transcription was significantly increased (Fig. 4D). However, ICRF193, a small catalytic inhibitor of TOP2, decreased the transcription (Fig. 4D), as previously shown. These results, together with the cryo-EM structure, strongly suggest that the DNA cleavage and bending caused by TOP2B positively regulate transcriptional activation of the EGR1 gene": First, etoposide and ICRF193 target both TOP2A and TOP2B. Thought unlikely, it's possible that the change of EGR1 transcription induced by etoposide and ICRF193 might be attributable to TOP2A. The authors should exclude this possibility. Second, the cryo-EM data do not establish that "DNA cleavage and bending caused by TOP2B positively regulate transcriptional activation of the EGR1 gene". It was already known that TOP2B bends DNA and is stabilized in a cleavage state by etoposide. This result may (or may not) be important for transcriptional activation but none of the experiments in the present work have any bearing on this hypothetical correlation.

Moreover, the EM data are at best of modest quality/resolution. The cryo-EM map of TOP2B seems to be stretched along the C2 symmetrical axis of TOP2B because of preferred orientation (Fig. 4). The actual quality of the cryo-EM map may be not as good as 3.9 Å. The authors may need to (a) try different conditions to get more orientations of the particles (supporting film like carbon/graphene/graphene oxide, detergent, tilting, crosslinking etc.) and/or (b) collect more cryo-EM data to increase the particle numbers. Only 26,067 particles were used for the final 3D reconstruction, which is probably too few to get a high-quality map of TOP2B.

Overall, there is no compelling reason for the EM findings to be included in the paper. They should be excised.

Minor points:

1) The authors may want add a schematic figure diagramming how transcriptional regulation of IEGs might occur through ERK2-topoisomerase II.

2) In Fig. 2, "ELKm" should be "ELK1m".

3) Fig. S1. The data do not demonstrate that ERK2 activates transcription (as the figure legend states). Rather, they all appear to be Western blots that comment only on recruitment. This

discrepancy should be corrected. Also, there is no test of ERK1 in lower part of panel B as the legend claims.

4) The lower panel of Supplementary Figure 1B seemed to be redundant.

5) Does "SCR" in Figure 5 mean "scrambled siRNA"? If yes, please explain in the figure legend.

6) In Fig. 5F and 5G, only the mRNA levels of EGR1/FOS for the ERK1 knockdown were shown. It would be useful to include TOP2B occupancies with the ERK1 knockdown study as well.

7) In Fig. 4D and Fig. 5, there are many different labels for the vertical axes (mRNA expression, Relative mRNA, and Relative mRNA expression). Is there any difference among them? If not, please use the same label for the vertical axes.

8) In Supplementary Figure 3:

(a) Panel A: It would be better to include a scale bar in the raw micrograph.

(b) "Toparz" should be "Topaz".

(c) A 3D FSC (<https://3dfsc.salk.edu/> or in Cryosparc) plot is a good supplement to the conventional FSC curve and local resolution map to present the quality of map.

(d) In Panel C, after 2D classification, the authors performed a first round of 3D classification but didn't throw away any bad classes. This is confusing.

(e) Imposing symmetry into the 3D refinement can sometimes introduce structural artifacts, especially with dynamic proteins. Imposing symmetry may also affect the conformation of DNA substrate in the TOP2B model. It would be more convincing to include a 3D reconstruction without imposing C2 symmetry in the supplementary material.

(f) The authors claim that "The density of the ATPase domain of TOP2B (1-456 aa) was observed in the 2D and 3D reconstructions (Supplementary Fig. 3D-F)". However, in Supplementary Fig. 3D-F, there was no density for the ATPase domains, and these regions were absent throughout 3D classification/refinement. Only in Supplementary Fig. 3B the blurred features for ATPase domains are evident. Please be more specific.

9) The paper states that "Both of the two DNA strands were cleaved at the active site of each TOP2B (Fig. 4B). Two etoposide molecules were intercalated into the two cleavage sites of the double strand of EGR1 TSS." However, the local density around the drug and DNA break is not shown. It is important to show the zoomed-in view of the density for this region in Supplementary Figure 3.

10) Per the instructions from PDB: "The preliminary report is not proof of deposition and should not be submitted to journals." The report produced at the annotation stage after the entry deposition is what should be submitted to the journal for manuscript review. The title page of this report shows the PDB ID, title, and deposition date, and includes a pink diagonal watermark "For Manuscript Review" on every page.

11) In Fig. 3B, why do the authors see a doublet of TOP2B? Is this due to a difference in phosphorylation status?

12) In Fig. 3G, which phosphorylation sites are already present in TOP2B prior to kinase treatment? It would be helpful to show them with the sites phosphorylated by ERK1/ERK2.

13) In Fig. S2A, the ELK1 preps don't appear terribly clean. Please comment.

Reviewer #3 (Remarks to the Author):

Kim et al., report a study of ERK2-TOP in transcription of immediate early genes. Overall, the manuscript is of broad interest and well written.

- I cannot find a description of the Supplementary Data files. Which Data file corresponds to which kinase reaction?

- Phosphorylation site localization scoring and/or manual inspection of the spectra would help determine if the assigned site is correct. This is particularly important for the conclusion that ERK1 and ERK2 have some unique phosphorylation target sites. Additionally, single replicate analysis of the phosphorylation sites maybe misleading in terms of identifying unique sites. Both these points raise the question of for kinase unique sites that are within a couple amino acids of sites phosphorylated by both kinases (e.g., 1342. Without multiple replicate analysis and phosphorylation site localization it is premature to determine these are really kinase unique sites.

- Mutational analysis and characterization of these ERK2 unique sites would support the speculation that the lack of their phosphorylation accounts for altered TOP2B accumulation. Perhaps this is beyond the scope of this study though.

- In several places % homology is written (E.g., on Page 3, 12). I think either % similarity or identity is meant. Sequences are either homologous (common evolutionary ancestry) or not.

- Depositing the raw spectra files at a public repository such as MassIVE or Pride would be beneficial.

Reviewers' comments:

Reviewer #1 (Remarks to the Author):

The manuscript by Kim et al. describes the potential role of human topoisomerase II β isoform (TOP2B) and of the phosphorylation of its C-terminal domain by the MAP kinase ERK2 in the regulation of immediate early gene transcription (EGR1). Inhibition or knock-down of ERK2 indicate that ERK2 activates transcription by increasing TOP2B binding to the transcription start site of EGR1 and FOS1.

→ We thank Reviewer 1 for his or her comments. However, we respectfully point out that the part shaded in yellow is less accurate. ERK2 catalytic inhibition or knock-out abnormally increased TOP2B binding to the tested genes, which suggested that ERK2 is necessary for proper TOP2B localization and catalysis. This was written in pages 15 and 17 of the original manuscript as below:

Page 15: Collectively, the results from ERK2 catalytic inhibition by VX11 and KD experiments suggest that the absence of functional ERK2 results in abnormal TOP2B accumulation, repressing transcription at *EGR1* gene.

Page 17: Consistently, ERK2 KD, but not ERK1 or double KD, increased TOP2B in the TSSs of *EGR1* and *FOS* genes (Figs. 5F,G). It is unclear what is the role of the accumulated TOP2B, an important question to be answered in future. Perhaps, these are the TOP2B molecules that might be either abnormally recruited or trapped (unable to be released from) on the DNA. TOP2B recruitment and catalysis of DNA double-strand breaks typically occur in the TSS of IEGs, leading to IEG activation (Figs. 3A, 5B)^{4,5}. However, in the case of ERK2 catalytic inhibition and KD, IEG expression was hindered (Figs. 5A–C, 5E–G), suggesting that TOP2B is not activated to catalyze the DNA but stays on the DNA in the absence of functional ERK2. This is supported by the observations that ICRF193, which causes TOP2B to be stuck on DNA without catalysis^{83,84}, inhibits transcription at *EGR1* and other IEGs (Fig. 4D)⁵.

The phosphorylation analysis of TOP2B brings interesting information but some experiments need to be better explained and justified. Together with the cellular assays, which results are more convincing, this study would benefit from further biochemical analysis to clarify the role and regulation of TOP2B activities by the phosphorylation.

→ We thank Reviewer 1 for his or her interest and suggestions. We made extensive efforts to fulfill the biochemical analyses suggested by Reviewer 1 in the revised manuscript.

In addition to in vitro and cellular assays, the authors present a cryoEM reconstruction of the TOP2B structure (DNA binding/cleavage domain). However, the structural data as presented does not support the conclusions on the role of TOP2B. This part would require more in-depth analysis and higher resolution structures if achievable, to provide structural details on the specificities of the EGR1 TSS DNA template compared with previous structures. The structure of this domain alone is not sufficient to account for the multiple interactions occurring during the assembly of large transcription complexes.

Many of the data presented in this article would need a thorough overhaul and reorganization of the figures to clarify the conclusions of the study.

→ We appreciate Reviewer 1's comments. We also would like to point out that the DNA binding domain of TOP2 and its associated DNA have been very difficult to obtain in crystal or cryoEM structures. For example, these regions could not be obtained in a high resolution in the most recent TOP2A structure (published in N. Communications in 2021)-- they had to reconstruct these regions through in silico prediction and biochemical data (personal communication with the corresponding author of the paper). This difficulty was stated with references in our original manuscript (pages 3 and 4). We believe that our TOP2B-*EGR1* TSS-etoposide cryoEM structure is comparable and absolutely informative and insightful because it is the first cryoEM structure of human TOP2B purified from human cells and it is in a complex with the actual, screened *EGR1* TSS sequence, not artificial sequences. The etoposide was mapped clearly in this near-native structure as well. Our TOP2B cryoEM structure indicates a similarity to the TOP2A one published in 2021.

In addition, we don't argue that the TOP2B-mediated bending of DNA is specific only to *EGR1* TSS because we did not compare it with other DNA segments. We believe that our results, obtained collectively from biochemical, cell-based, mass spectrometry, and structural analyses, suggest that *EGR1* gene transcription is activated by ERK2, that TOP2B is regulated by ERK2 for proper engagement with *EGR1* (and *FOS*), and that TOP2B-mediated DNA break and bending of the *EGR1* TSS is a positive element for gene activation.

Specific comments and questions:

In vitro transcription experiments

- The authors are incubating the *EGR1* DNA template with a HeLa nuclear extract to compare the levels of transcription activation upon addition of ERK1 and ERK2. What are the levels of ERK1 and ERK2 endogenous proteins for activated transcription compared with the in vitro assay? What are the levels of ERK1/2 in the HeLa nuclear extract relative to the amount of recombinant ERK1/2 added in the assay?

→ We thank Reviewer 1 for these questions. The amount of ERK proteins added to the immobilized template assay was approximately 100 ng per each reaction. For reviewer's questions, we quantified the endogenous ERK1 and ERK2 proteins in HeLa NE as well as the template bound ERKs. Although the quantification can give only rough ideas due to presumably different antibody sensitivity and quality etc between ERK1 and ERK2 antibodies, ERK2 appeared more abundant than ERK1 in HeLa NE (see INPUT of fig. 1E in the revised manuscript). The different nucleus entry rates between ERK1 and ERK2 were in fact reported previously (Marchi, M. et al. 2008; Whitehurst, A.W. et al. 2002). Both endogenous ERK1 and ERK2 in NE were inefficient in binding to the template (see CTRL in fig. 1E). Even 30 min development of the immunoblots couldn't get any signal for ERK1 and ERK2, suggesting that the ERK1 and ERK2 in NE may not be free (or blocked) to associate with the template. This result also supports a previous finding that ERK2 is tethered in nucleus (Caunt, C. J. et al. 2012; Mandl, M. et al. 2005). On the other hand, the recombinant ERK1 and ERK2, when supplemented, were associated with the template. Therefore, we think that the background activities by endogenous ERK proteins are probably negligent- attributing the transcription factor recruitment to the activities of recombinant ERK proteins in our immobilized template assays. We also note that the

ERK1 antibody seemed to cross-react a bit to ERK2 while the ERK2 antibody was very specific only to ERK2.

- Figure 1: Panel C displays histograms for the levels of proteins associated with DNA template assessed by immunoblotting (WB shown in supplementary figure 1). On the contrary, panel E shows the immunoblotting of the proteins. The figures (main and supplementary) should be reorganized. The conclusions drawn from these experiments do not seem completely supported by the data. In particular the levels of CDK9 or MED23 do not seem significantly higher than in the control experiments in panel E, or in supplementary 1 panel B.

→ To enhance the visualization and statistical power, we changed the immunoblots into graphs in Fig. 1F and modified Fig. S1 in the revised manuscript. CDK9 and MED23 are increased by ERK2 and ERK2m and the increase is statistically significant.

- Could the authors explain how the value of enrichment is assessed for the "none" condition on panel 1C, to which other control or basal level is it compared ?

→ We thank Reviewer 1 for this comment. In Fig. 1C, "none" was referred to just kinase (ERK)-free, buffer only supplement (ERK storage buffer), a negative control. For more straightforward comparisons, the enrichment was shown in fold change (the buffer control as 1) in the revised manuscript.

- Panel 1F: the authors claim that ERK1 has an opposite effect than ERK2 but the EGR1 transcript level for ERK1 is within the same range as control level. It looks like ERK1 has no effect when compared to ERK2 and it seems difficult to conclude at this stage that they have an "opposite" effect based on this particular experiment.

→ We respectively point out that in Fig. 1F, ERK1 or ERK1m reduced *EGR1* transcription *in vitro* significantly (P values < 0.005 and < 0.0005).

- Figure 2/3: the organization of data in supplementary figure 1 relative to the main figure 2/3 is confusing. Supplementary figure 1B is cited numerous times and it seems like some panels could move in the main figures.

→ We rearranged supplementary figure 1A, B, C and tried to avoid multiple citing in the revised manuscript.

- The use of "m" for either a point mutation in the protein or a mutation in the DNA sequence is also confusing. The terms "CTRL", "NONE" and "WT" refer to different controls in each panel and should be more clearly explained in the figure legends.

→ We thank Reviewer 1 for this comment. We have changed the terms for easier recognition in the revised manuscript. ELKm was changed to be ELK1^{mut} and Buffer only controls (CTRL, None, WT etc) were to CTRL throughout the manuscript.

- The authors claim that TOP2B association with EGR1 is positively correlated with ELK1 binding but

this binding may be due also to the fact that Pol II has a deficient binding on ELK1m template, since TOP2B has been shown to associate with Pol II complexes.

→ We appreciate Reviewer 1's valuable discussion. It is possible that TOP2B recruitment is compromised on the ELK1m template through direct (from ERK or ELK interaction to recruit TOP2B) and/or indirect (through other proteins such as Pol II) interactions. However, we note that we did not observe a significant, consistent reduction of Pol II binding to the ELK1m template unless ERKs were supplemented (Figs. 2B, D in the original manuscript). On the other hand, TOP2B was reduced on the mutant template with or without ERKs.

- Supplementary figure 2A: could the authors explain the presence of 2 bands for the ELK1 recombinant sample ? these 2 bands do not appear in figure 3D and the single band in figure 3D appears between the 37 and 50 kDa markers where there is no significant protein band in supplementary figure 2A.

→ The purified ELK1 proteins show two major bands in SDS-PAGE at around 37 and 60 KDa and both of them were confirmed to be ELK1 by Western blotting (Supplementary Figure 2A). We note that ELK1 contains large unstructured regions and therefore the recombinant protein is very sensitive to proteases. Although it is less compelling, the protein purity is competitive as ELK1 is a challenging protein to purify from *E. coli* (commercial ELK1 by Abcam shown below). Please note that our gel showing ELK1 in Fig. S2A was stained with silver (detecting proteins < 2 ng) while the Abcam commercial ELK1 with Coomassie blue. In Fig. 3C (silver staining) and 3D (autoradiography), we see these two ELK1 species are phosphorylated by ERK2 at a bit above 37 KDa and around 60 KDa (see below for WB data for ELK1 at around 60 KDa). Therefore, the brackets were extended accordingly in the revised manuscript.

[REDACTED]

[REDACTED]

- Supplementary figure 2B: the legend should explain that the image is a ³²P autoradiography of a SDS-PAGE gel.

→ We appreciate Reviewer 1 for this suggestion. The description has been included in Supplementary Figure 2B.

- Figure 3D: the intensity of TOP2B phosphorylation by WT ERKs is indeed very low, the authors should provide another image with only the WT ERKs to be able to compare with the control (no kinase) ? What is the kinase/TOP2B ratio and would the signal be higher with an increased ratio ? The ERKs can indeed be used to identify more robustly the phosphorylation sites, but it seems important to measure the effect of the wt ERKs rather than of the ERKs to assess the effect of the kinases on the TOP2B.

→ TOP2B only control was shown in Fig. 3D. It is less surprising that, without upstream activators in the *in vitro* kinase assay (unlike immobilized template assay with HeLa NE), WT ERKs do not robustly phosphorylate TOP2B but mildly as shown in Fig. 3D. The titration assay with ERKs was already included in Supplementary Fig. 2B. We tested ERKs with other substrates in the same conditions as with TOP2B and found that ERKs specifically and strongly phosphorylates TOP2B. The data are yet confidential that we cannot include them in the manuscript. We assure Reviewer 1 that the TOP2B phosphorylation by constitutively active ERKs results from specific enzyme-substrate interaction. The ratio was described in Materials and Methods and Supplementary Fig. 2B in the original manuscript as below:

“The substrate, 1.8 μg of TOP2B was incubated with approximately 0.25–0.5 μg WT or mutant ERK proteins as a kinase in a reaction. For the control reaction, TOP2B without any kinase was prepared side-by-side.”

“Titration of ERK1s and ERK2s in the *in vitro* kinase assay showing a dose-dependent increase of phosphorylation of TOP2B. Nine μg of TOP2B and 0, 0.25, 0.5 and 1 μg of ERK1s or ERK2s were incubated for 1 h at 30 °C.”

These ratios have been ones typically used as standards in the *in vitro* kinase assays and accepted/published in many previous studies (Stein et al. JBC 1996; New et al. EMBO J. 1998;

Bancerek et al. Immunity 2013).

Structural data processing

The data processing is insufficiently described.

- Particle picking: what was the rationale behind the use of 2 different particle picking softwares ? How many particles were identified simultaneously by the 2 softwares ? Could the authors explain how the particles were then extracted and pulled ? How was the first 3D reconstruction obtained, in particular what is the ab-initio model if there is one?

→ We appreciate the Reviewer's questions. We used both the warp and topaz softwares to maximize particle numbers. The particles picked by the two softwares were merged, and overlapping particles were removed as shown in Supplementary Fig. 3. The particles were extracted by using the Relion program. The first 3D map was reconstructed by using the Relion 3D initial model.

- Initial and final models for the DNA binding/cleavage region:

The authors used the Alphafold2 model for Top2B as an initial template for model building. Which regions were used to start the model building, provided that some domains are disordered and not properly modelled? (domain boundaries ?)

What is the completeness of the model compared to the crystallographic structures of TOP2B DNA binding/cleavage domain? How does the final cryoEM model compares with the AlphaFold model in term of completeness and RMSD ?

→ We used the Alphafold2 model of full-length TOP2B as an initial template. Subsequently, disordered regions were removed from model. The final model includes the regions of 445–1118 and 1140-1201 amino-acid residues. The crystal structure of TOP2B-DNA-etoposide complex (PDBID: 3QX3) includes 452-696, 706-1110, and 1135-1201 amino-acid residues. RMSD between the two models is ~1.75 Å.

- Although weaker, the density for the ATPase domain is visible in some of the 2D classes. The authors should consider running 3D classification without C2 symmetry to attempt to see this region. Running of a non-uniform refinement and/or homogeneous refinement in cryoSPARC with and without the C2 symmetry should be done to attempt to improve the resolution.

→ We appreciate the Reviewer's constructive suggestions. We had tried non-uniform and homogeneous refinements in cryoSPARC with and without the C2 symmetry. However, the map quality is still not sufficient for model building of the ATPase domain, probably due to the small size of this domain. This result was described in the Methods section.

DNA: 3D classification without alignment with a mask focus on DNA region should be tried to improve density of DNA and see if the DNA sequence can be identified or if the density of DNA base pairs corresponds to an average density?

→ We appreciate the Reviewer's constructive suggestions. 3D classification without alignment with a mask focus on DNA region was performed, but we could not identify the DNA sequence. It should be noted that there is no sequence specificity in DNA cleavage by TOP2B.

- Structural analysis:

- There is no figure showing the density at the site of cleavage supporting the claim that DNA is cleaved or that etoposide molecules are bound.

→ The figure of the density around the etoposide molecules and the DNA cleavage sites is now provided in Supplementary Figs. 5A,B.

How many base pairs are visible in the density out of the 50 bp used as DNA template?

- As mentioned by the authors, several structures of the homologous human TOP2 have been solved, either individual domains using xray-crystallography or full-length proteins using cryoEM (ref 52, 53, 54). All structures show sharp bending of the DNA molecule and some also have doubly nicked DNA with etoposide molecules inserted at the cleavage sites.

The DNA bending and cleavage mechanism is an intrinsic mechanism of the Top2, regardless of the DNA sequence. Is the orientation and bending of the EGR1 TSS template differing from the published structures, and if so, how? Is there any particular geometry in this promoter region that would support the claim that this is how transcription activation is happening? Superimposition of structures and indication of the bending angles would be useful.

If no structural difference, is there any difference on the flexibility/molecular dynamics of this DNA sequence?

→ We sincerely appreciate Reviewer 1 for this discussion and suggestion. Here are some facts to be considered and compared (see the table below). Ref 52 is the cryo-EM of hTOP2A, not hTOP2B. Indeed, DNA bending appears similar in these two isoforms. Ref 53 is indeed for hTOP2B, a crystal structure, not cryo-EM one. It was purified from *E. coli* and the DNA segment is only 20 bp—DNA bending is less obviously depicted in their model (see below for the comparison), compared to our hTOP2B cryoEM structure, which shows almost 90 degree bending. Ref 54 used the core domains of hTOP2B and 20 bp short DNA for crystal structure. In this 2011 paper (Ref 54), DNA bending is also less pronounced (see below for the comparison). In addition, even if DNA bending and cleavage is what is expected for how topoisomerases works, our hTOP2B was purified from human cells and its cryo-EM structure shows the DNA bending clearly, for the first time, which is novel and important to be published. Although we cannot argue that this bending is specific/unique to *EGR1* TSS because we did not compare different DNA segments, the bending of *EGR1* TSS by TOP2B catalysis is also note-worthy.

For Reviewer's suggestion, the indication of the bending angles has been newly included in the revised manuscript.

The *EGR1* TSS, the 50 bp TSS sequence of *EGR1* was chosen because this segment showed a higher affinity to TOP2B in EMSA experiments. The data have been newly included in the Supplementary Figure 3. Given that our study is focused on TOP2B regulation in *EGR1* gene, the structural data propose that DNA cleavage and dramatic bending by TOP2B occurs in the *EGR1* TSS for gene activation.

Article Year	Our manuscript 2022	Ref 53 (N. Commun) 2021	Ref 54 (N. Commun) 2018	Ref 55 (Science) 2011

TOP2/DNA bp	hTOP2B /50 bp EGR1 TSS	hTOP2A /30 bp random sequence	hTOP2B /20 bp random sequence	hTOP2B /20 bp random sequence
Method	Cryo-EM	Cryo-EM	Crystal	Crystal
Structure				Resolution	3.9 Å	3.6–7.4 Å	2.74 Å	2.16 Å

- The structural data do not seem to explain why treatment of HEK293 cells by etoposide shows increased EGR1 transcription, and why treatment by ICRF-193 (targeting the ATPase domain, not visible in the cryoEM density) shows decreased transcription. DNA bending and cleavage occur without etoposide.

→ Our cell-based experiments showed that etoposide activates *EGR1* transcription, but ICRF-193 suppresses it. Etoposide extends the duration cleavage and therefore bending, compared to TOP2 by itself because it inhibits the instant resealing of the broken DNA strands. In fact, it is more widely accepted that TOP2-mediated DNA cleavage is instantaneously and spontaneously rejoined by the same enzyme. On the other hand, ICRF-193 reduces the ability of TOP2B to cleave DNA and therefore bending because it is a catalytic inhibitor as reviewer mentions. Therefore, there is an apparent correlation between DNA break and gene activation. We suggest that DNA bending and break, which is visualized by TOP2B-*EGR1* TSS-etoposide complex (Figs. 4A–C), explains how etoposide treatment increases *EGR1* transcription (Fig. 4D).

- The authors could not reconstruct density for the C-terminal region that is bearing most of the phosphorylation sites, which makes sense since this region is disordered. However, 2 sites common to ERK1 and 2 are located on the DNA binding/cleavage core that was reconstructed (Figure 3G). An overall 3.9 Å resolution is not high enough to distinguish phosphate groups but local resolution may be better on some regions. In the TOP2B protein produced in HEK cells, without in vitro phosphorylation, are these positions phosphorylated? Is it possible to distinguish phosphorylation on positions T715 or S728? On which sub-domains are these residues positioned? Which positions are crucial for TOP2B recruitment on the EGR1 TSS? To probe this model, it would

be interesting to analyze point mutations on phosphorylated positions specific to ERK2 in the CTD in the cellular assays, and their impact on the catalytic activities.

→ We appreciate Reviewer 1 for these comments. Because the TOP2B protein used for *in vitro* kinase assay and mass spectrometry was purified from HEK cells, we found that it was already extensively phosphorylated in many residues. TOP2B samples were re-examined for multiple times by mass spectrometry and the background phosphorylated residues appear heterogenous too. To circumvent this background issue, in the preparation of revision, we purified TOP2B while stripping phosphates from the protein by using phosphatases. This dephosphorylated TOP2B (deTOP2B) was used for *in vitro* kinase assay and mass spectrometry. As shown in the Supplementary Data 1, the new control showed less extensive phosphorylation compared to the native one, and ERK1/ERK2 mutual and ERK2 specific residues were identified. The data with either native and deTOP2B pointed out ERK1- and ERK2-mediated phosphorylation mostly at the TOP2B CTD. deTOP2B experiments suggested four ERK2-specific residues and couples of ERK1/ERK2 mutual residues. Mutational study, combined with *in vitro* and *in vivo* functional analyses, will be important to validate the phospho-sites suggested by this study. Because these analyses require extensive sets of assays, we respectfully request Reviewer 1's understanding that they will be ideal for the further study.

Reviewer #2 (Remarks to the Author):

In the present manuscript, Kim et al. report that ERK2, but not ERK1, is important for IEG transcriptional activation. They also identify a possible ELK1 binding site on the EGR1 gene that appears to be important for ERK2 transactivation at EGR1 and the enrichment of TOP2B on the EGR1 transcription start site (TSS). The CTD of TOP2B is noted to be extensively phosphorylated by both ERK1 and ERK2, while the failure of this phosphorylation of TOP2B is linked to the accumulation of TOP2B on TSS of IEGs.

The organization and findings of the manuscript are not always logically laid out and were difficult to follow at times. The primary conclusion of this work proposes that TOP2B association, catalysis, and dissociation on DNA are important for regulating transcription and that ERK2-mediated TOP2B phosphorylation may be important for this process. Although there are new findings that are interesting (such as TOP2B being a substrate for ERK1/2), many of the data are insufficient and at times inconsistent to support the stated claims.

→ We thank Reviewer 2 for his or her interest in our findings and critical suggestions to improve the manuscript. We have tried our best effort to address Reviewer 2's comments and suggestions in the revised manuscript and sincerely ask that reviewer can appreciate it.

Primary comments

1) There are number of issues concerning various aspects of the recruitment and transcriptional activation data. For example,

– In Fig. 1E, the constitutively active mutant ERK2m does not appear to be more proficient in recruiting Pol II, MED23 and CDK9 than WT ERK2; however, in Fig. 1F the ERK2m activated transcription more efficiently than ERK2 WT. This dichotomy should be explained.

→ In Fig. 1E and Supplementary Fig. 2B of the original manuscript, the Pol II and transcription activators, MED23 and CDK9 appear slightly more recruited by ERK2m than by ERK2. In Fig. 1F, *EGR1* transcription is increased 2 fold for ERK2 and about 3 fold for ERK2m in average, thus 50% increase between these two. Therefore, the results appear consistent between Fig. 1E and 1F. To enhance the visibility and statistical power, the previous Fig. 1E was substituted to graphs including statistical validations in the revised manuscript (Fig 1F).

– In Fig. S1, the effects of recruitment by the various ERK constructs seem inconsistent. For example, S1A shows a slight increase for CDK9 with ERK2, while Fig. 1E does not. By comparison, Fig. S1B shows no effect on MED23 or TOP2B recruitment by any of the ERKs. Also, why does ERK1 appear to antagonize MED23 recruitment on the mutant ELK1 sequence? Finally, why does ERK2 appear to stimulate the recruitment of MED23 in panel A, but have no effect in panel B? The inconsistencies between replicates are problematic.

→ As mentioned above, to circumvent the misinterpretation and confusion, we substitute Fig. 1E to graphs and reorganized the supplementary figure 1. We show that CD9 is increased by both ERK2 and ERK2m.

If Reviewer 2 means panel A and B for Supplementary Fig. 1A and 1B, these results were obtained from different experimental conditions. In Supplementary Fig. 1A, without NTP data need to be compared with Supplementary Fig. 1B, which are consistent. To circumvent the confusion, we have reorganized the supplementary Fig. 1 in the revised manuscript.

– In Fig. 5E the intensity of ERK1/2 in K1KD and K2KD are significantly different. This raises a concern that ERK1 or ERK2 may not be knocked down efficiently. The authors should check this with a specific antibody for ERK1 and ERK2, respectively.

→ We appreciate Reviewer 2's suggestion. ERK1 and ERK2 protein expression levels appear different in HEK293 cells as Fig. 5E shows. In addition, ERK1 and ERK2 are very similar in their sizes 43 KDa and 41 KDa, respectively. Depending on the cell lines and their phosphorylation status in cells, their sizes are also different from the expected sizes (see the attached figure at the bottom). Depending on the cell lines, they can appear even as one band (perhaps for PTMs)... and we could not find any reference showing ERKs immunoblotting in HEK293 cells.

[REDACTED]

These facts contribute to the difficulty to visualize ERK1 and ERK2 in HEK293 cells. For Reviewer 2's suggestion, ERK1 and ERK2 antibodies were used to show the KD efficiency and the results have been included in Fig. 5E in the revised manuscript.

– In Fig. 2B, the signal for Pol II seemed to be stronger in the ELK1m group than WT. Is this assay performed without adding ERK proteins? If yes, then why in Supplementary Figure 1B is the signal for Pol II much weaker in the ELK1m group than in the WT?

→ We respectfully disagree. In Fig. 2B (without ERK included), it is impossible to argue that Pol II signal is stronger in ELK1m, either by eyes or by Image J software. In the original Fig. S1B, Pol II signal is also almost similar without ERKs supplement. However, with ERKs supplement, yes, ELK1m template showed weaker Pol II signal than WT template did.

And in Fig. 2C, why should the decrease noted for TOP2B be more statistically significant than the decrease reported for ELK1 and CDK9, when the degree of the decrease is smaller for TOP2B than for the other two proteins?

→ We respectfully remind Reviewer 2 that the argument, the degree of decrease/increase determines statistical significance, is scientifically/statistically unsupported. In addition, both $P < 0.005$ (ELK1 and CDK9) and $P < 0.0005$ (TOP2B) are considered to be statistically significant.

– The data shown in Fig. 5F-G do not comport with the authors' claims. If the presence of ERK2 has a positive role on gene expression and a negative impact on TOP2B accumulation (for *EGR1*), then this should be seen with the double knockout (which looks like WT). These data also appear to run counter to those shown for Fig. 3A, where ERK2 is claimed to boost TOP2B recruitment.

→ ERK2 inhibition thus shows a mild, yet statistically consistent decreased *EGR1* mRNA expression (Fig. 5F). It is not surprising that the double KD without both ERK1 and ERK2 is similar to WT because both ERK1-mediated inhibition and ERK2-mediated activation are void in the cell (Fig. 5F). ERK2 activates *EGR1* transcription in which TOP2B is recruited to the TSS of *EGR1* gene (Fig. 3A). However, without ERK2, TOP2B recruitment or dissociation is strikingly abnormal (Figs. 5D, F, G). Interestingly, ERK1 KD decreases TOP2B in the *EGR1* and *FOS* TSS (Fig. 5F, newly included), which may explain why the double KD doesn't affect TOP2B occupancy at these genes significantly.

Again, ERK2 enhances TOP2B recruitment upon *EGR1* activation. However, without ERK2, TOP2B occupancy is abnormally increased. We discussed this as below in our original manuscript:

Page 15: Collectively, the results from ERK2 catalytic inhibition by VX11 and KD experiments suggest that the absence of functional ERK2 results in abnormal TOP2B accumulation, repressing transcription at *EGR1* gene.

Page 17: Consistently, ERK2 KD, but not ERK1 or double KD, increased TOP2B in the TSSs of *EGR1* and *FOS* genes (Figs. 5F,G). It is unclear what is the role of the accumulated TOP2B, an important question to be answered in future. Perhaps, these are the TOP2B molecules that might be either abnormally recruited or trapped (unable to be released from) on the DNA. TOP2B recruitment and catalysis of DNA double-strand breaks typically occur in the TSS of IEGs, leading to IEG activation (Figs. 3A, 5B)^{4,5}. However, in the case of ERK2 catalytic inhibition and KD, IEG expression was hindered (Figs. 5A–C, 5E–G), suggesting that TOP2B is not activated to catalyze the DNA but stays on the DNA in the absence of functional ERK2.

– Figs. 1F, 2D, 2E, 3A, 4D, 5A-D, and 5F-G: the authors need to show examples of the raw data used to generate bar graphs, as well as more clearly point out which data are representative of each graph.

→ In our previous experience, Nature Communications require most raw data for the main figures in publication. We followed our internal rules as: 1. Statistical representations were presented for the data that were produced using multiple biological and/or technical replicates. 2. If more than two consecutive experiments showed qualitatively similar results, a representative image was displayed. 3. Based on sound scientific rationales and ethics, figures and graphs were generated and presented. For Reviewer 2's suggestion, figure presentation has been further clarified in the revised manuscript.

Overall, the inconsistencies in the data and/or their presentation make it difficult to reconcile many of the observed findings with the stated claims.

→ We think that there was some misunderstanding as we explained above. To enhance the comprehensibility, we included additional data and reorganized the figures in the revised manuscript.

2) In Fig. S2B, the kinase data are worrisome. Kinases are typically robust enzymes. The authors need 0.5-1 μg of kinase to see a signal on 9 μg of substrate after a full hour of incubation. This result indicates that the specific activity of the kinases is quite low, which raises concerns about their use and specificity in other experiments.

→ In fact, for Fig. S2B, the autoradiograms were taken less than 0.5 min because the signals were strong. In addition, we incubated the samples at 20 °C (RT) or 30 °C, not typical 37 °C that most kinase assays reported. In Fig. S2B, the dose-dependent (0.25 – 1 μg) increase of phosphorylation represents the specificity. In addition, we compared TOP2B phosphorylation by ERK with another protein side-by-side. Although the data cannot be included in the manuscript (preparing another manuscript with the data), these controls and comparisons strongly suggested the specific enzyme-substrate interaction between ERK and TOP2B. In addition, WT ERKs without upstream MAPK proteins served as controls for constitutively active ERKs that do not require upstream activators, to validate the specificity. In Fig. 3D and mass spectrometry analyses, 1.8 μg of TOP2B and about 200 ng of ERK were used. For immobilized template assay, about 100 ng of ERK per a reaction was supplemented.

We would like to remind Reviewer 2 that the kinase and substrate amounts and incubation time used for this study are within standards (Stein et al. JBC 1996; New et al. EMBO J. 1998; Zhou et al. Mol. Cell 2017; Bancerek et al. Immunity 2013 etc.). Furthermore, we validated the specific interaction and identified the mutual and distinctive ERK-mediated TOP2B phosphorylation sites by mass spectrometry analyses.

3) In Fig. 3C, the band of TOP2B in the lane TOP2B/K1 looks quite different from the bands in other lanes. Are there controls to ensure that the same amount of protein was loaded into all the wells?

→ As Reviewer 2 probably knows, the silver-stained gel in Fig. 3C and the Coomassie blue-stained gel in Fig. 3E show the reactions after *in vitro* kinase reaction. The amounts/protein concentrations of proteins used for the assay were presented in the Materials & Methods and shown in silver stained gels. Protein concentrations were calculated using Bradford Assays and confirmed by the silver-stained gels. The equal amounts of substrates and kinases were used for the kinase assay and an equal amount of reactions was loaded onto the SDS-PAGE gel. Pre-kinase assay proteins were shown in Figs. 1B, 1D, 3B, and S2A.

4) Regarding the statement “However, in the case of ERK2 catalytic inhibition and KD, IEG expression was hindered (Figs. 5A–C, 5E–G), suggesting that TOP2B is not activated to catalyze the DNA but stays on the DNA in the absence of functional ERK2” (Discussion section): ERK2 has many substrates besides TOP2B, including Pol II and ELK1. Thus, VX11 treatment or knock down of ERK2 would change the phosphorylation state of various proteins, which may lead to a change in IEGs transcription and TOP2B occupancy at the TSS.

→ We appreciate Reviewer 2 for the discussion. Although we showed the interaction between ERK2 and TOP2B *in vitro*, the aberrant increase of TOP2B without functional ERK2 could result from direct

and indirect interactions among multiple transcription factors and Pol II as Reviewer 2 pointed out. This has been discussed in the revised manuscript.

Based on the data in Fig. 5 it is hard to tell whether TOP2B is indeed “not activated to catalyze the DNA but stays on the DNA in the absence of functional ERK2”. The authors should compare the relaxation/cleavage activity of TOP2B with/without phosphorylation by ERK2 (and ERK1) *in vitro*. If TOP2B becomes more active following ERK2 treatment, then it might be more reasonable to make this claim.

→ The data in Fig. 5 suggested that TOP2B, in spite of abnormal increase in its occupancy is not activated to catalyze DNA/induce DSB without functional ERK2 because TOP2B-mediated DSB by etoposide activated EGR1 transcription. In *in vitro* conditions, TOP2B catalyzes supercoiled substrates too well even without ERKs or any specific modifications. We ask Reviewer to understand the dilemma to recapitulate the *in vivo* conditions, in which TOP2B-DNA interaction and catalysis should be regulated, *in vitro*.

5) The authors showed that both ERK1 and ERK2 could phosphorylate TOP2B, and most of their phospho-sites overlap. By contrast, ERK1 and ERK2 showed differential effects on EGR1 transcription. The authors should explain why the phosphorylation of residues 1342, 1403, 1473, and 1588 led to different effect of ERK2 from ERK1.

→ In our original manuscript, we used HEK-purified TOP2B, which was extensively phosphorylated, for the *in vitro* kinase assays. The background phosphorylation in the control inhibited to fairly screen the phosphorylation sites. In preparation of the revision, we tried to circumvent this problem: TOP2B was dephosphorylated using phosphatases in the purification process and the dephosphorylated TOP2B was used for the *in vitro* kinase assay, followed by mass spectrometry. These efforts could improve the background issue and suggested a set of ERK1 and ERK2 mutual phospho-sites and ERK2-specific sites in the TOP2B CTD.

6) Only the 5' end of one strand of the DNA template used in the immobilized template assay was fixed by attaching to the beads, which means that the DNA template will always be relaxed without any supercoiling tension during transcription. The supercoiled state of DNA template may affect transcription, raising concerns that the enrichment of TOP2B and Pol II, CDK9, and MED23 in Fig. 1-3 may not be physiologically relevant. Please comment.

→ We appreciate Reviewer 2 for this inquiry. In fact, there are strengths and weaknesses for the *in vitro* immobilized template and transcription assays, just like all other experiments. For most *in vitro* immobilized template assays, naked DNA is used as the template DNA and the topology might not be quite the same as what is in the cell, as one can imagine. However, when we did mass spectrometry and immunoblotting analyses with different template DNAs [HSP70 (Bunch et al. NSMB 2014) and EGR1 (Bunch et al. Open Biology 2021) fragments], general transcription and pausing factors were highly enriched in the TSS of these templates than the control/mutant fragments (GREB1, HSP70 HSF1/TATA mutant templates). In addition, certain proteins that are less likely to be on the naked DNA template such as histone proteins and helicases were identified as abundant by mass spectrometry. Therefore, the protein assembly in the *in vitro* assays has been proven to be reliable and informative even if it may not be the exactly same as it is in the cell. The immobilized template

assay combined with transcription assay are classical and powerful *in vitro* biochemical methods to understand transcription, specifically the function of transcription factors (<https://www.mbi.ucla.edu/archives/faculty/michael-carey>). The weakness of cell-based analyses could be that the conditions are rather undefined, while the strength of the immobilized template/transcription assays is that they can be conducted in more controlled manners. Therefore, we combined these biochemical and cell-based analyses as well as structural analyses to validate the hypothesis and results and to suggest a novel finding, based on the comprehensive data, in this study.

7) Regarding the statement “When HEK293 cells were treated with 10 μ M etoposide for 3 h, EGR1 transcription was significantly increased (Fig. 4D). However, ICRF193, a small catalytic inhibitor of TOP2, decreased the transcription (Fig. 4D), as previously shown. These results, together with the cryo-EM structure, strongly suggest that the DNA cleavage and bending caused by TOP2B positively regulate transcriptional activation of the EGR1 gene”: First, etoposide and ICRF193 target both TOP2A and TOP2B. Thought unlikely, it’s possible that the change of EGR1 transcription induced by etoposide and ICRF193 might be attributable to TOP2A. The authors should exclude this possibility.

→ As Reviewer 2 mentioned, it is possible that TOP2A might be also regulated by ERK2 and could be an important activator in *EGR1* transcription as a recent study indicated (Herrero-Ruiz et al. Cell Rep. 2021). However, this study did not include or test TOP2A and it is difficult to suggest anything without the entire set of rigorous experiments with TOP2A. Because our study has been focused on TOP2B function in transcription since 2014 and this study is mainly to understand the relation between ERK and TOP2B, we think that TOP2A is out of this study’s scope but is definitely an important future study. We have newly included this possibility in the Discussion section in the revised manuscript.

Second, the cryo-EM data do not establish that “DNA cleavage and bending caused by TOP2B positively regulate transcriptional activation of the EGR1 gene”. It was already known that TOP2B bends DNA and is stabilized in a cleavage state by etoposide. This result may (or may not) be important for transcriptional activation but none of the experiments in the present work have any bearing on this hypothetical correlation.

→ We respectfully request Reviewer 2 to understand that our cryo-EM data combined with cell-based analyses data are important to suggest DNA cleavage and bending caused by TOP2B positively regulate transcriptional activation of the *EGR1* gene. As we presented a comparison among the human TOP2 structures below, in spite of observed similarities, our cryo-EM structure of TOP2B-*EGR1* TSS-etoposide is distinctive and novel from previous ones in that:

1. A full-length TOP2B (not TOP2A) purified from human cells (not from bacterial cells) was visualized for the first time
2. Not a random DNA sequence but the *EGR1* segment with a higher affinity to TOP2B was complexed with TOP2B
3. Importantly, the degree of DNA bending in TOP2B was relatively more clearly captured in our structure and appears comparable to the TOP2A cryo-EM structure published in 2021.

Article	Our manuscript	Ref 52 (N. Commun)	Ref 53 (N. Commun)	Ref 54 (Science)
---------	----------------	--------------------	--------------------	------------------

Year	2022	2021	2018	2011
TOP2/DNA bp	hTOP2B/50 bp EGR1 TSS	hTOP2A/30 bp random sequence	hTOP2B/20 bp random sequence	hTOP2B/20 bp random sequence
Method	Cryo-EM	Cryo-EM	Crystal	Crystal
Structure				Resolution	3.9 Å	3.6–7.4 Å	2.74 Å	2.16 Å

We showed that etoposide treatment increased *EGR1* expression but ICRF193 did not (Fig. 4D): this side-by-side comparison is also novel and noteworthy. We attempted without being successful to obtain the cryo-EM structure of TOP2B-*EGR1* TSS-ICRF193. However, it is noted that the ICRF193 complex structure appeared much different from TOP2B-*EGR1* TSS-etoposide. Nonetheless, DNA cleavage and dramatic bending captured by etoposide have been shown (Figs. 4A-C), in which *EGR1* expression is activated (Fig. 4D).

Moreover, the EM data are at best of modest quality/resolution. The cryo-EM map of TOP2B seems to be stretched along the C2 symmetrical axis of TOP2B because of preferred orientation (Fig. 4). The actual quality of the cryo-EM map may be not as good as 3.9 Å. The authors may need to (a) try different conditions to get more orientations of the particles (supporting film like carbon/graphene/graphene oxide, detergent, tilting, crosslinking etc.) and/or (b) collect more cryo-EM data to increase the particle numbers. Only 26,067 particles were used for the final 3D reconstruction, which is probably too few to get a high-quality map of TOP2B.

Overall, there is no compelling reason for the EM findings to be included in the paper. They should be excised.

→ We respectfully request Reviewer 2 to appreciate our cryo-EM structure for the reasons described above. In addition, as the comparison table shows, the recent TOP2A structure published in Nature Communications (2021) was 3.6–7.4 Å with the challenging domains constructed by in silico predictions and simulations from biochemical data (personal conversation with the corresponding author of the paper). We believe that our structural resolution is competitive and novel as it is the first cryo-EM structure of human TOP2B purified from human cells.

Minor points:

1) The authors may want add a schematic figure diagramming how transcriptional regulation of IEGs might occur through ERK2-topoisomerase II.

→ We thank Reviewer 2 for this suggestion. A schematic representation (Fig. 6) has been newly included in the revised manuscript.

2) In Fig. 2, “ELKm” should be “ELK1m”.

→ For reviewers’ suggestions, ELK1m has been changed to ELK1^{mut} throughout the revised manuscript.

3) Fig. S1. The data do not demonstrate that ERK2 activates transcription (as the figure legend states). Rather, they all appear to be Western blots that comment only on recruitment. This discrepancy should be corrected. Also, there is no test of ERK1 in lower part of panel B as the legend claims.

→ Fig. S1 has been rearranged and revised.

4) The lower panel of Supplementary Figure 1B seemed to be redundant.

→ Supplementary Figure 1B has been reorganized for Reviewer’s suggestion.

5) Does “SCR” in Figure 5 mean “scrambled siRNA”? If yes, please explain in the figure legend.

→ We thank Reviewer 2 for this comment. SCR in Fig. 5 was explained in the legend of the revised manuscript.

6) In Fig. 5F and 5G, only the mRNA levels of EGR1/FOS for the ERK1 knockdown were shown. It would be useful to include TOP2B occupancies with the ERK1 knockdown study as well.

→ ChIP-qPCR data showing TOP2B occupancy change in ERK1 KD have been newly included in the revised manuscript.

7) In Fig. 4D and Fig. 5, there are many different labels for the vertical axes (mRNA expression, Relative mRNA, and Relative mRNA expression). Is there any difference among them? If not, please use the same label for the vertical axes.

→ We thank Reviewer 2 for this suggestion. These different labels for the Y axis of qRT-PCR data have been changed to mRNA expression throughout the manuscript.

8) In Supplementary Figure 3:

(a) Panel A: It would be better to include a scale bar in the raw micrograph.

→ A scale bar is now added in the Supplementary Fig. 4A.

(b) “Toparz” should be “Topaz”.

→ The typo has been corrected.

(c) A 3D FSC (<https://3dfsc.salk.edu/> or in Cryosparc) plot is a good supplement to the conventional FSC curve and local resolution map to present the quality of map.

→ We generated a 3D FSC plot, which is now included in Supplementary Fig. 4G.

(d) In Panel C, after 2D classification, the authors performed a first round of 3D classification but didn't throw away any bad classes. This is confusing.

→ We corrected the flowchart of Panel C.

(e) Imposing symmetry into the 3D refinement can sometimes introduce structural artifacts, especially with dynamic proteins. Imposing symmetry may also affect the conformation of DNA substrate in the TOP2B model. It would be more convincing to include a 3D reconstruction without imposing C2 symmetry in the supplementary material.

→ We appreciate the Reviewer's constructive comments. We performed 3D reconstruction without imposing C2 symmetry, and the map is now shown in Supplementary Fig. 5E. There is no large difference between the densities with and without imposing the C2 symmetry. This result was described in the Methods section.

(f) The authors claim that “The density of the ATPase domain of TOP2B (1–456 aa) was observed in the 2D and 3D reconstructions (Supplementary Fig. 3D–F).” However, in Supplementary Fig. 3D–F, there was no density for the ATPase domains, and these regions were absent throughout 3D classification/refinement. Only in Supplementary Fig. 3B the blurred features for ATPase domains are evident. Please be more specific.

→ If the contour level is lowered, the density for the ATPase domain is obscurely observed. Our 3D refinement using the mask focused on the ATPase domain did not yield high-quality map of this domain, probably because of its small size and flexibility. Therefore, we did not build a model of this domain. This point was described in the Methods section.

9) The paper states that “Both of the two DNA strands were cleaved at the active site of each TOP2B (Fig. 4B). Two etoposide molecules were intercalated into the two cleavage sites of the double strand of EGR1 TSS.” However, the local density around the drug and DNA break is not shown. It is important to show the zoomed-in view of the density for this region in Supplementary Figure 3.

→ We appreciate the Reviewer's important advice. The figure of the density around the etoposide molecules and the DNA cleavage sites is now provided in Supplementary Figs. 5A,B.

10) Per the instructions from PDB: “The preliminary report is not proof of deposition and should not be submitted to journals.” The report produced at the annotation stage after the entry deposition is what should be submitted to the journal for manuscript review. The title page of this report shows the PDB ID, title, and deposition date, and includes a pink diagonal watermark “For Manuscript Review” on every page.

→ The validation report for manuscript review is attached to the revised manuscript.

11) In Fig. 3B, why do the authors see a doublet of TOP2B? Is this due to a difference in phosphorylation status?

→ Although it could be TOP2B species with different PTMs for the bands, we think that it is more likely to be a smear of the protein due to the slight conformational differences (partial folding, for example) in the low gel percentage part of a gradient gel. Please see the 250 KDa standard protein right next to the TOP2B protein, which also shows a doublet. In addition, our manual silver stain protocol can detect proteins < 2 ng, very sensitive and it might have detected a small quantity of proteins that migrated slightly different for the partial folding. As you know, silver staining is not quantitative like Coomassie staining. Please note that this smeared band for TOP2B was unseen in Fig. 3E or other Coomassie staining.

12) In Fig. 3G, which phosphorylation sites are already present in TOP2B prior to kinase treatment? It would be helpful to show them with the sites phosphorylated by ERK1/ERK2.

→ This information has been included in the Supplementary Data 1.

13) In Fig. S2A, the ELK1 preps don't appear terribly clean. Please comment.

→ As Reviewer 2 mentions, we acknowledge that the ELK1 prep was less satisfactory to us. However, it should be noted that we tried multiple different preps, following previous references with or without modifications and can say that ELK1 protein purification from *E. coli* is quite challenging. In spite, we think that our bacterial purified ELK1 protein is in relatively good quality even with silver staining. To compare, please see the figure below showing a commercial ELK1 (Abcam) purified from *E. coli* and stained with Coomassie. In addition, our ELK1 protein was verified by immunoblotting, and worked well as a ERK2 catalytic substrate.

[REDACTED]

Reviewer #3 (Remarks to the Author):

Kim et al., report a study of ERK2-TOP in transcription of immediate early genes. Overall, the manuscript is of broad interest and well written.

→ We sincerely thank Reviewer 3 for his or her positive reading of our manuscript.

- I cannot find a description of the Supplementary Data files. Which Data file corresponds to which kinase reaction?

→ The description of Supplementary Data files has been included in the supplementary information of the revised manuscript.

- Phosphorylation site localization scoring and/or manual inspection of the spectra would help determine if the assigned site is correct. This is particularly important for the conclusion that ERK1 and ERK2 have some unique phosphorylation target sites.

→ We appreciate this suggestion. The manual inspection of the spectra (Supplementary Data 2) has been included in the revised manuscript.

Additionally, single replicate analysis of the phosphorylation sites maybe misleading in terms of identifying unique sites. Both these points raise the question of for kinase unique sites that are within a couple amino acids of sites phosphorylated by both kinases (e.g., 1342. Without multiple replicate analysis and phosphorylation site localization it is premature to determine these are really kinase unique sites.

→ We thank Reviewer 3 for this suggestion. The TOP2B protein (native TOP2B) was heavily phosphorylated when it was purified in HEK cells, causing the background signal issues. In fact, the phosphorylation of native TOP2B proteins appeared heterogenous even in a single protein prep too. These made the identification of ERK-mediated phosphorylation sites challenging in multiple attempts. Therefore, during the revision preparation, we dephosphorylated TOP2B by stripping phosphates by using phosphatases in protein purification steps (deTOP2B). Although deTOP2B still showed phosphorylated residues, the number was much less. The assays with deTOP2B suggested ERK1/2 mutual sites and ERK2-specific sites. As Reviewer 3 mentioned, these sites, mutual and specific, are close by one another, which poses ambiguity and caution in some degree. However, ERK1- and ERK2 mutual sites as well as ERK2 specific sites have been identified extensively in the TOP2B CTS, with either native TOP2B or deTOP2B. We confidently suggest that ERK1 and ERK2 phosphorylate mainly the TOP2B CTD.

We believe and mentioned in the Discussion section that the phospho-sites suggested by this study should be validated by mutational study, combined with *in vitro* and *in vivo* functional analyses in future.

- Mutational analysis and characterization of these ERK2 unique sites would support the speculation

that the lack of their phosphorylation accounts for altered TOP2B accumulation. Perhaps this is beyond the scope of this study though.

→ As discussed above, the point mutation study of ERK2 specific phosphorylation sites requires a series of rigorous experiments as these sites can show effects as a single mutation or a combination of multiple mutations. We think that the mutational analysis of these phosphorylation sites will be interesting and important for the further study.

- In several places % homology is written (E.g., on Page 3, 12). I think either % similarity or identity is meant. Sequences are either homologous (common evolutionary ancestry) or not.

→ We appreciate this suggestion. Percent homology on pages 3 and 12 was changed to % identity in 3 places.

- Depositing the raw spectra files at a public repository such as MassIVE or Pride would be beneficial.

→ The raw files have been deposited onto jPOST, a member of the Proteome Xchange consortium (<https://www.proteomexchange.org/>), in which MassIVE and PRIDE also participate. The accession numbers are PXD040977 for ProteomeXchange and JPST002096 for jPOST. Currently they are registered in private and we will open them to the public manually when the paper becomes published.

URL <https://repository.jpostdb.org/preview/4741084146418eddfa8391>
Access key 4661 (Available until announcement)

Reviewers' comments:

Reviewer #1 (Remarks to the Author):

The revised manuscript by Kim et al. describes the role of human topoisomerase II β isoform (TOP2B) and of the phosphorylation of its C-terminal domain by the MAP kinase ERK2 in the regulation of immediate early gene transcription (EGR1).

The authors have amended the manuscript and added some clarifications and controls as requested. The manuscript and the figures are now clearer.

The authors wanted to get structural information for the molecular details of TOP2B interaction with the IEG sequence. Despite the authors explanations, it is still difficult to see how the structure explains the effects of the promoter.

It is well documented that structures of hsTOP2 are difficult to obtain. Compared to other hs TOP2B structures, the DNA in this structure is longer, confirming bending. DNA bending however happens on any sequence in type 2A topoisomerases.

If anything, this structure proves that TOP2B can bind and cleave this DNA sequence, but it could also with other DNA sequences.

According to previous biochemistry data (and new supplementary figure 3), TOP2B may bind potentially with higher affinity to this promoter, which may be the most important point to convey here, also in the abstract.

For the readers of this manuscript, it is important to clearly state that there is no molecular evidence in the present structural data of the DNA binding-cleavage domain alone, to explain the specificity for this particular sequence.

In the discussion, the following sentence should be also nuanced: "TOP2B catalyzes the DNA to produce DNA break that generates DNA bending (Fig. 4) and provokes the activation of phosphoinositide 3 (PI3) kinases⁴⁻⁶"

DNA bending within this domain happens as soon as bound to TOP2, before, or without cleavage as shown in other type 2A structures with un-cleaved DNA sequences, or even inactive tyrosine mutants. Bending is necessary for the hydroxyl group of the catalytic tyrosine to reach the phosphate backbone and generate cleavage. It is likely that bending is happening first in the sequence of events.

Comments on supplementary figure 3:

The authors don't explain how they deduce from 3B that one sequence is a stronger binder to TOP2B? Is it the disappearance of the (DNA?) band at the 1:2 ratio?

Since this is an important point in the study, this should be explained or the gel should be annotated on the side for the presence/absence of a band corresponding or not, to the formation of the complex.

Also if the gel was silver-stained, this should be mentioned in the legend to understand what is shown on the gel, even if it's the same conditions as in reference 6.

Reviewer #2 (Remarks to the Author):

The authors addressed many minor concerns but did not satisfactorily resolve several significant comments/points raised during the prior review.

1. In revision to Fig. 5E, ERK1 seems to be knocked out completely in the K1KD group, but in the K2KD group the amount of remaining ERK2 is high. The authors' response focuses on how difficult it will be to immunoblot ERK1 and ERK2 in HEK293 cells, but this is irrelevant to the original question.

2. The original point raised that in "Figs. 1F, 2D, 2E, 3A, 4D, 5A-D, and 5F-G: the authors need to show examples of the raw data used to generate bar graphs, as well as more clearly point out which data are representative of each graph" is not addressed in the response but still needs to be. The authors state that "figure presentation has been further clarified in the revised manuscript", but raw data are still not provided in the revised manuscript.

3. One of the previous comments noted during the first review mentioned that "Based on the data in Fig. 5 it is hard to tell whether TOP2B is indeed "not activated to catalyze the DNA but stays on the DNA in the absence of functional ERK2". The authors should compare the relaxation/cleavage activity of TOP2B with/without phosphorylation by ERK2 (and ERK1) in vitro. If TOP2B becomes more active following ERK2 treatment, then it might be more reasonable to make this claim." This issue remains unaddressed in the revision. The response to the comment also doesn't make sense. Even if ERK2 is knocked out, TOP2B itself should still be active to cut the DNA and religate the DNA. The data in Fig. 5 indicate that the knockdown of ERK2 reduces the transcription level of EGR1 and increase TOP2B occupancy; however, from these data, it is unclear how one can conclude that "TOP2B is not activated to catalyze the DNA " in the absence of functional ERK2? The original comment recommended that the authors perform an in vitro activity assay of TOP2B with or without ERK2 phosphorylation, but the authors ask me to understand the dilemma to recapitulate the in vivo conditions. Why does the in vitro assay need to recapitulate the in vivo conditions?

4. As requested, a figure was added showing the local density around the etoposide and the DNA cleavage site. However, the resolution of these regions appears too low to properly fit an etoposide molecule or model the DNA cleavage site.

5. The response claiming that cryo-EM structure is important, novel, of high quality, and should remain in the paper is unconvincing:

a) Although the TOP2B enzyme imaged in this work does correspond to the full-length protein purified from human cells, the resultant structure provides no new information about the physical organization of the enzyme or its catalytic mechanism (e.g., how binds/cleaves DNA, how etoposide works, etc.). Without such insights, it is highly derivative and of marginal significance to the field.

b) The authors attempt to link the extent of DNA bending seen in the structure with the activation of gene expression. However, no structural or biochemical evidence is provided to support this claim. The degree of bending shown in the figures in the paper is comparable to that seen in other TOP2 structures, and there are no new interactions shown between the protein and the DNA or drug that might be informative mechanistically. The inability to make any such inference or connection again indicates that the structural work simply reinforces what is already known in the field.

c) The response claims that the TOP2B structure presented here is of similar quality to a recent TOP2A structure from the Lamour group. However, the structure showcased in the present work has a strong preferred orientation (as shown by the 3DFSC) that significantly reduces map quality; by comparison, the TOP2A structural work does not suffer from such problems and has would appear to have better map quality overall. The TOP2B cryo-EM maps shown in the revision still need to be improved and used to either: 1) answer a question pertinent to the ERK perspective of the paper or 2) highlight some new aspect of topoisomerase and/or etoposide function. Without such insights, the structural studies remain of limited value that detract from the primary theme of the paper and should be excised.

Reviewer #3 (Remarks to the Author):

Thank you for addressing my comments.

Reviewers' comments:

Reviewer #1 (Remarks to the Author):

The revised manuscript by Kim et al. describes the role of human topoisomerase II β isoform (TOP2B) and of the phosphorylation of its C-terminal domain by the MAP kinase ERK2 in the regulation of immediate early gene transcription (EGR1).

The authors have amended the manuscript and added some clarifications and controls as requested. The manuscript and the figures are now clearer.

→ We appreciate Reviewer 1 for the positive comments.

The authors wanted to get structural information for the molecular details of TOP2B interaction with the IEG sequence. Despite the authors explanations, it is still difficult to see how the structure explains the effects of the promoter.

It is well documented that structures of hsTOP2 are difficult to obtain. Compared to other hs TOP2B structures, the DNA in this structure is longer, confirming bending. DNA bending however happens on any sequence in type 2A topoisomerases.

If anything, this structure proves that TOP2B can bind and cleave this DNA sequence, but it could also with other DNA sequences.

According to previous biochemistry data (and new supplementary figure 3), TOP2B may bind potentially with higher affinity to this promoter, which may be the most important point to convey here, also in the abstract.

For the readers of this manuscript, it is important to clearly state that there is no molecular evidence in the present structural data of the DNA binding-cleavage domain alone, to explain the specificity for this particular sequence.

→ We thank Reviewer 1 for the constructive suggestions. The suggested information about the DNA of the TOP2B structure and the limitation of specificity have been newly included in the revised manuscript.

In the discussion, the following sentence should be also nuanced : “ TOP2B catalyzes the DNA to produce DNA break that generates DNA bending (Fig. 4) and provokes the activation of phosphoinositide 3 (PI3) kinases4-6”

DNA bending within this domain happens as soon as bound to TOP2, before, or without cleavage as shown in other type 2A structures with un-cleaved DNA sequences, or even inactive tyrosine mutants. Bending is necessary for the hydroxyl group of the catalytic tyrosine to reach the phosphate backbone and generate cleavage. It is likely that bending is happening first in the sequence of events.

→ We appreciate Reviewer 1 for these helpful discussions. The sentence has been revised.

Comments on supplementary figure 3:

The authors don't explain how they deduce from 3B that one sequence is a stronger binder to TOP2B ? is it the disappearance of the (DNA?) band at the 1:2 ratio ?

Since this is an important point in the study, this should be explained or the gel should be annotated on the side for the presence/absence of a band corresponding or not, to the formation of the complex.

Also if the gel was silver-stained, this should be mentioned in the legend to understand what is shown on the gel, even if it's the same conditions as in reference 6.

→ We thank Reviewer 1 for this request and acknowledge that it is important to clarify the Supplementary Figure 3. Initially, we did not elaborate the findings in this figure because they have been already published in our previous paper (*Bunch et al. Open Biology 2021*). In the revised manuscript, the gels have been annotated with free DNA vs DNA-TOP2B complex and the staining (using silver nitrate) has been mentioned in the figure legend. In addition, a graph with the averages and standard deviations from two independent assays has been newly included to exhibit the strongest interaction between EGR1 #3-3 (within -423 to +332 of *EGR1* gene) and TOP2B.

Once more, we appreciate Reviewer 1 for his or her constructive comments and time to review our manuscript.

Reviewer #2 (Remarks to the Author):

The authors addressed many minor concerns but did not satisfactorily resolve several significant comments/points raised during the prior review.

1. In revision to Fig. 5E, ERK1 seems to be knocked out completely in the K1KD group, but in the K2KD group the amount of remaining ERK2 is high. The authors' response focuses on how difficult it will be to immunoblot ERK1 and ERK2 in HEK293 cells, but this is irrelevant to the original question.

→ Previously we used an ERK1/ERK2 antibody from Abcam and it was difficult to detect both proteins well for their similar molecular weights (42 and 44 KDa) and one of the bands were more intense to shadow the other band. As Reviewer 2 requested in the previous comments, we have used the separate ERK1 and ERK2 antibodies from Santa Cruz Biotechnology and Abcam and these antibodies could detect the proteins well. We appreciate Reviewer 2 for this. Although ERK2 KD efficiency appeared rather low, the KD showed statistically-significant effects in the tested assays.

2. The original point raised that in “Figs. 1F, 2D, 2E, 3A, 4D, 5A-D, and 5F-G: the authors need to show examples of the raw data used to generate bar graphs, as well as more clearly point out which data are representative of each graph” is not addressed in the response but still needs to be. The authors state that “figure presentation has been further clarified in the revised manuscript”, but raw data are still not provided in the revised manuscript.

→ The representative raw data requested by Reviewer 2 have been enclosed in the revision submission.

3. One of the previous comments noted during the first review mentioned that “Based on the data in Fig. 5 it is hard to tell whether TOP2B is indeed “not activated to catalyze the DNA but stays on the DNA in the absence of functional ERK2”. The authors should compare the relaxation/cleavage activity of TOP2B with/without phosphorylation by ERK2 (and ERK1) *in vitro*. If TOP2B becomes more active following ERK2 treatment, then it might be more reasonable to make this claim.” This issue remains unaddressed in the revision. The response to the comment also doesn't make sense. Even if ERK2 is knocked out, TOP2B itself should still be active to cut the DNA and religate the DNA. The data in Fig. 5 indicate that the knockdown of ERK2 reduces the transcription level of EGR1 and increase TOP2B occupancy; however, from these data, it is unclear how one can conclude that “TOP2B is not activated to catalyze the DNA ” in the absence of functional ERK2? The original comment recommended that the authors perform an *in vitro* activity assay of TOP2B with or without ERK2 phosphorylation, but the authors ask me to understand the dilemma to recapitulate the *in vivo* conditions. Why does the *in vitro* assay need to recapitulate the *in vivo* conditions?

→ For the previous revision, we followed the typical DNA relaxation assay protocols (e.g. ~30 min incubation) to compare TOP2B catalytic activity with or without ERKs and could not detect notable differences. This was probably because TOP2B is a proficient and very efficient enzyme to instantaneously relax supercoiled DNA by itself *in vitro*. In addition, we could not (and still cannot) exclude the possibility that ERK2-mediated TOP2B phosphorylation indirectly modulates TOP2B affinity with DNA or TOP2B catalysis through allowing/excluding other proteins to interact with TOP2B. These were why we asked Reviewer 2's understanding for the limitation of *in vitro* activity assays.

To address Reviewer 2's suggestion, we performed the *in vitro* DNA relaxation assay using mildly-modified protocols for this revision. For Pol II pause release, it is hypothesized that positive supercoiling ahead of paused Pol II needs to be relaxed by topoisomerases. Therefore, we purchased validated positively supercoiled pBR322 from a UK-based company (Inspiralis) to test TOP2B activity to relax the DNA. For the typical 30 min incubation, there was no difference to detect in TOP2B catalysis with or without ERKs (**Fig. 5C**). However, for short 6 min incubation and low TOP2B concentrations (5, 50 nM), we could see the effects of ERK1- and ERK2-mediated augmentation of TOP2B catalysis (**Fig. 5A**). TOP2B was a bit more effective to relax the DNA in the presence of ERK2 than ERK1 although the difference was seen when TOP2B concentrations were low and the reaction time was short. In dynamic and robust transcriptional activation which has to occur within mins in the case of IEGs, ERK2-mediated enhancement of TOP2B catalysis to relax the positive supercoiling might be critical.

Importantly, the relaxation assays suggested a new function of ERK1 that the constitutively active ERK1 (ERK1m) might be capable of relaxing the positively supercoiled DNA and it also might modulate DNA conformation (**Fig. 5D**). In addition, we found that ERK1m generated a unique-sized band, from the size of the band, suggesting a semi-relaxed form of the positively supercoiled DNA substrate, and this DNA was resistant to be relaxed by TOP2B (**Figs. 5C,D**). Although further experiments are needed, these data suggest that ERK1 might repress gene expression at *EGR1* and *FOS* genes through altering DNA topology that hinders TOP2B catalysis and thus transcription. Rigorous biochemical and molecular biology experiments will be necessary to fully understand this potentially important, new function of ERK1 in future.

ERK2 KD and chemical inhibition increased TOP2B occupancies at the *EGR1* and *FOS* TSSs. This could be due to the decreased enzyme rate without ERK2, stalling TOP2B on the DNA longer. Alternatively, as mentioned above, ERK2-mediated TOP2B phosphorylation may have indirect effects through regulating other players that interact with TOP2B in the cell. These proteins could be important for, for example, releasing TOP2B from the DNA and TOP2B catalysis.

4. As requested, a figure was added showing the local density around the etoposide and the DNA cleavage site. However, the resolution of these regions appears too low to properly fit an etoposide molecule or model the DNA cleavage site.

→ The map quality of the current TOP2B structure is insufficient to resolve atomic details. This may be partly because the current complex contains the full-length, highly phosphorylated TOP2B and a longer stretch of DNA. However, while we have avoided detailed structural description in the revised manuscript, the current resolution is sufficient to discuss whether or not the overall structures of TOP2B and DNA are similar between the current and previous structures. This is the first report of a cryoEM structure for TOP2B and demonstrates that the full-length recombinant TOP2B expressed in human cells with a phosphorylated C-terminal domain and a DNA substrate can be visualized by cryoEM. Comparable to the TOP2A cryoEM structure, the ATPase domain is not well resolved nor is the CTD well resolved, but the core breakage-reunion domain is resolved to 3.9 Å showing that this region, within the full length, post translationally modified enzyme, is directly comparable to the crystal structure of the TOP2B core crystallized with recombinant protein expressed in bacteria, this is important information.

5. The response claiming that cryo-EM structure is important, novel, of high quality, and should remain in the paper is unconvincing:

a) Although the TOP2B enzyme imaged in this work does correspond to the full-length protein purified from human cells, the resultant structure provides no new information about the physical organization of the enzyme or its catalytic mechanism (e.g., how binds/cleaves DNA, how etoposide works, etc.). Without such insights, it is highly derivative and of marginal significance to the field.

→ We present the first cryo-EM structure of TOP2B, using full-length recombinant enzyme purified from human cells, thus post-transcriptionally modified enzyme. It clearly demonstrates that the core domain forms a similar structure to the TOP2B core crystallized with recombinant protein expressed in bacteria. The TOP2B is complexed to a biologically relevant DNA segment within the *EGR1* gene, a high affinity binding site for TOP2B within –432 to +332 of the *EGR1* gene, this DNA-protein complex is stabilized with TOP2 poison etoposide. Although the overall structure is similar to TOP2A or previously published crystal structures of TOP2B, that doesn't invalidate or void the firstness of our structure.

b) The authors attempt to link the extent of DNA bending seen in the structure with the activation of gene expression. However, no structural or biochemical evidence is provided to support this claim. The degree of bending shown in the figures in the paper is comparable to that seen in other TOP2 structures, and there are no new interactions shown between the protein and the DNA or drug that might be informative mechanistically. The inability to make any such inference or connection again indicates that the structural work simply reinforces what is already known in the field.

c) The response claims that the TOP2B structure presented here is of similar quality to a recent TOP2A structure from the Lamour group. However, the structure showcased in the present work has a strong preferred orientation (as shown by the 3DFSC) that significantly reduces map quality; by comparison, the TOP2A structural work does not suffer from such problems and has would appear to have better map quality overall. The TOP2B cryo-EM maps shown in the revision still need to be improved and used to either: 1) answer a question pertinent to the ERK perspective of the paper or 2) highlight some new aspect of topoisomerase and/or etoposide function. Without such insights, the structural studies remain of limited value that detract from the primary theme of the paper and should be excised.

→ We appreciate this reviewer's constructive comment. As written above, the electron density of the current TOP2B structure is insufficient to resolve atomic details. However, the current resolution is sufficient to compare overall structures of the current full-length, phosphorylated TOP2B with the previous structures. In the revised manuscript, we have focused on describing what was observed in the TOP2B structure, and have avoided discussing the link between the DNA bending and gene activation. The current structure contains a longer DNA stretch compared to those in the previous TOP2B structures, and we observed the TOP2B interaction with the distal DNA regions (**Supplementary Fig. 6**), though a similar interaction was also observed in the TOP2A structure. Nonetheless, the current structure may support that the presence of CTD and its extensive phosphorylation does not have a large influence on the core structure of TOP2B and its DNA-binding mode. This result supports the idea that the CTD phosphorylation may either facilitate the TOP2B recruitment/release to/from the *EGR1* DNA or indirectly activate the TOP2B enzymatic activity, for example, by removing autoinhibition (Jeong *et al. Elife* 2022, PMID: 36342377).

Reviewer #3 (Remarks to the Author):

Thank you for addressing my comments.

→ You are welcome and we sincerely appreciate your review.

REVIEWER COMMENTS

Reviewer #2 (Remarks to the Author):

The second revision addressed some issues, but more concerns were introduced, especially the presentation and interpretation of DNA topology data. These issues significantly damage the credibility of this manuscript, so they must be addressed before publication.

- (a). The stop solution (40% glycerol, 100 mM Tris-HCl (pH 8.0), 1 mM EDTA, 0.5 mg/mL bromophenol blue) is doubtful to quench the reaction. The final concentration of EDTA would only be 0.5 mM, while the concentration of Mg²⁺ would be 5 mM. So, there is 10-fold less EDTA present than what would be needed to quench the reaction.
- (b). The caption of Figure 5A said "Both ERK1/1m and ERK2/2m enhance the catalytic activity of TOP2B, compared to the CTRL, while ERK2 and ERK2m stimulate TOP2B catalysis more than ERK1 and ERK1m at 5 and 50 nM of TOP2B (purple arrows for comparison)", which is not true as the control experiment was very poorly done. The gel image shows a much fainter supercoiled band in the presence of 5 nM TOP2B compared to 0 nM and 50 nM TOP2B. The graph at the bottom of Fig. 5A doesn't match what is seen in the gel for the control relaxation assay at all.
- (c). In Figure 5A, I didn't see the unique DNA band marked with the red arrow. The quality of the gel is too poor, and the background is too high.
- (d). The purchased positively supercoiled plasmid is not pure, and the upper band should be nicked plasmid, but the authors labeled it as relaxed plasmid in Figure 5A.
- (e). In Figure 5C, how do the authors know that the band marked with the red arrow is "semi-relaxed" form of DNA? It looks like a linearized form of DNA to me. Rigorous control experiments need to be done to confirm that the band is real "semi-relaxed" or linearized (such as running the plasmid linearized by restriction endonuclease as control).
- (f). What's more important, the authors further concluded that "ERK1m can relax the DNA by itself and it forms the semi-relaxed DNA (Fig. 5D). The gradual accumulation of semi-relaxed DNA band and DNA relaxation were observed in both 10 and 30 min reactions, with or without ATP. (Fig. 5D)." This claim is not solid, as it's still unknown whether the so-called semi-relaxed DNA band is linearized. Therefore, it's also possible that the ERK1m preparation is contaminated by ATP-independent DNase. The authors should exclude this possibility before claiming that ERK1m can relax positively supercoiled plasmid.
- (g). The RNA polymerase will overtwist DNA downstream and undertwist DNA upstream, which means the DNA behind RNAP will be negatively supercoiled. (Reference to PMID: 28275417) In Figure 7, the TOP2B seems to be behind Pol II, but in the caption of Figure 7 it said "TOP2B generates DNA bending, relaxes DNA positive supercoiling". Here the positive supercoiling should be negative supercoiling. Accordingly, it will be more relevant to test the relaxation activity of TOP2B in the presence of ERK with negatively supercoiled plasmid.
- (h). The whole paragraph on Page 20 highlighted by cyan color is overinterpretation. There is no quantification to back the claim. Also, the bands they are referring to have the same migration pattern on the EtBr-free gel, so they don't appear to be of different topological states.

REVIEWER COMMENTS

Reviewer #2 (Remarks to the Author):

The second revision addressed some issues, but more concerns were introduced, especially the presentation and interpretation of DNA topology data. These issues significantly damage the credibility of this manuscript, so they must be addressed before publication.

(a). The stop solution (40% glycerol, 100 mM Tris-HCl (pH 8.0), 1 mM EDTA, 0.5 mg/mL bromophenol blue) is doubtful to quench the reaction. The final concentration of EDTA would only be 0.5 mM, while the concentration of Mg²⁺ would be 5 mM. So, there is 10-fold less EDTA present than what would be needed to quench the reaction.

→ We appreciate Reviewer 2 for this helpful comment. The relaxation assay buffers were prepared according to the manufacturer's instruction provided by the company. For the reviewer's comment, we contacted the company (Inspiralis) with this issue. The company admitted this mistake in their protocol and acknowledged that the STOP buffer should include 10 mM EDTA. However, along with the STOP buffer, two other measures were simultaneously applied to stop the reaction: the samples were treated with cold PCI and put on ice immediately after the addition of the cold STOP buffer. The company mentioned in email that these applications certainly stop the reaction. In spite, we used the STOP buffer including 10 mM EDTA in the revision work and revised the description the materials and methods section to be clearer: along with the STOP solution, cold PCI was simultaneously added to the reaction on ice.

(b). The caption of Figure 5A said "Both ERK1/1m and ERK2/2m enhance the catalytic activity of TOP2B, compared to the CTRL, while ERK2 and ERK2m stimulate TOP2B catalysis more than ERK1 and ERK1m at 5 and 50 nM of TOP2B (purple arrows for comparison)", which is not true as the control experiment was very poorly done. The gel image shows a much fainter supercoiled band in the presence of 5 nM TOP2B compared to 0 nM and 50 nM TOP2B. The graph at the bottom of Fig. 5A doesn't match what is seen in the gel for the control relaxation assay at all.

→ The graph and gel image of Fig. 5A have been revised.

(c). In Figure 5A, I didn't see the unique DNA band marked with the red arrow. The quality of the gel is too poor, and the background is too high.

→ The arrow was removed in the revised manuscript.

(d). The purchased positively supercoiled plasmid is not pure, and the upper band should be nicked plasmid, but the authors labeled it as relaxed plasmid in Figure 5A.

→ We thank Reviewer 2 for the comment. As the references below (published papers and company description) show, the upper bands can include both nicked and relaxed

DNA. The label for the upper band in Fig. 5 has been revised and labeled as open circular (OC) and relaxed plasmid in the revised manuscript.

<https://www.inspiralis.com/technical/methods-overview/topo-iv-relaxation-assays/>

<https://doi.org/10.1073/pnas.1700721114>

<https://doi.org/10.7845/kjm.2020.9074>

<https://doi.org/10.1016/j.nucmedbio.2021.06.004>

[REDACTED]

(e). In Figure 5C, how do the authors know that the band marked with the red arrow is “semi-relaxed” form of DNA? It looks like a linearized form of DNA to me. Rigorous control experiments need to be done to confirm that the band is real “semi-relaxed” or linearized (such as running the plasmid linearized by restriction endonuclease as control).

→ We sincerely appreciate Reviewer 2 for this comment, which was important to improve the clarity of the findings. Originally, we discussed but dismissed the possibility that the arrowed band might be a linearized form of pBR322 because the size of the band appeared to be ~3000 bp (pBR322: 4361 bp). It was a mistake from misreading the size marker in our part. We repeated the experiments including a linearized plasmid (cut with HindIII) and found that the band generated by ERK1m runs close to the linearized one. In addition, we confirmed that the open circular/relaxed band is increased by the presence of ERK1m. Although we cannot determine and completely distinguish the linearized DNA from semi-relaxed one and the OC one from relaxed plasmids this time, the original and additional experiments together clarified that ERK1m can convert positively (and negatively) supercoiled pBR322 plasmid DNA (bottom band) into OC/linearized/relaxed one (upper bands). The manuscript has been revised accordingly. In addition, we will continuously study the unexpected ERK1m ability to relax the supercoiled DNA with different approaches in future.

(f). What’s more important, the authors further concluded that “ERK1m can relax the DNA by itself and it forms the semi-relaxed DNA (Fig. 5D). The gradual accumulation of semi-relaxed DNA band and DNA relaxation were observed in both 10 and 30 min reactions, with or without ATP. (Fig. 5D).” This claim is not solid, as it’s still unknown whether the so-called semi-relaxed DNA band is linearized. Therefore, it’s also possible that the ERK1m preparation is contaminated by ATP-independent DNase. The authors should exclude this possibility before claiming that ERK1m can relax positively supercoiled plasmid.

→ As described above, we have performed the repeated experiments including the linearized plasmid and the text has been carefully revised to be truthful to what is observed in the assay results. The additional data have been also included in the revised manuscript. In addition, we have investigated the suspected contamination of any DNase in ERK1m protein preps. We have performed multiple mass spectrometry (MS) analyses for this study and so the MS data were screened against *E. coli* proteome data banks to compare the CTRL, ERK1m and ERK2m samples and to identify any potential DNA stand breakers (any DNases, nucleases). The search did not identify plausible factors, specific to the ERK1m prep, that might linearize/relax the plasmid DNA (Supplementary Data 8–10). For example, there was single strand specific exonuclease in all three samples of CTRL, ERK1m, and ERK2m, in a very low score (16–17). Although the MS analysis is very sensitive, we acknowledge that there is always a possibility of some tiny proteins undiscovered in MS. In spite, we have little evidence to support the possibility of factor contamination from the current data sets acquired through silver staining of purified proteins, mass spectrometry analysis of in vitro reactions, and side-by-side ERK1-ERK2-ERK1m-ERK2m protein purification.

(g). The RNA polymerase will overtwist DNA downstream and undertwist DNA upstream, which means the DNA behind RNAP will be negatively supercoiled. (Reference to PMID: 28275417) In Figure 7, the TOP2B seems to be behind Pol II, but in the caption of Figure 7 it said “TOP2B generates DNA bending, relaxes DNA positive supercoiling”. Here the positive supercoiling should be negative supercoiling. Accordingly, it will be more relevant to test the relaxation activity of TOP2B in the presence of ERK with negatively supercoiled plasmid.

→ We thank Reviewer 2 for the discussion and suggestion. It is known that transcription induces negative supercoiling behind Pol II and positive supercoiling ahead of Pol II. Although Pol II is resilient to torsion/DNA supercoiling, both negative and positive supercoiling halt Pol II to be translocated (PMID: 23812716). In particular, in promoter-proximal Pol II pausing, the positive supercoiling that is likely formed ahead of paused Pol II (<https://doi.org/10.7554/eLife.67236>) could be reinforced by the first nucleosome and a lack of precedent Pol II in the gene body: it must be relaxed for active transcription. TOP2 can relax both positive and negative supercoiling and therefore, during gene activation, the function of TOP2 to resolve the positive and negative supercoiling is likely to be regulated for productive transcription. We initially tested the positive supercoiling for the reason. As the reference listed above showed, negative supercoiling, in excess, can halt Pol II. On the other hand, multiple studies (PMID: 1992462; PMID: 28275417) suggest that negative supercoiling can activate transcription by facilitating the melting of the DNA to be more accessible to Pol II and transcription factors. For the reviewer’s suggestion, we tested TOP2B catalysis to relax negatively supercoiled pBR322 plasmid (purchased from Inspiralis) in the presence of CTRL, ERK1m, and ERK2m. The analysis turned out to be extremely valuable, showing that ERK2m decreases TOP2B catalysis to relax negatively supercoiled DNA while ERK1m rarely affects it, compared to the CTRL. In addition, similarly to the positively supercoiled DNA, ERK1m relaxes/linearizes negative supercoiled pBR322 by itself. These data suggest differential effects of ERK1 and ERK2 to regulate TOP2B and that ERK2m may delay TOP2B catalysis of the negative supercoiling while promoting TOP2B to relax positive supercoiling for transcriptional activation. The manuscript has been revised with the new data and we will continuously work on this phenomenon to reveal the regulation and detailed mechanisms in future.

(h). The whole paragraph on Page 20 highlighted by cyan color is overinterpretation. There is no quantification to back the claim. Also, the bands they are referring to have the same migration pattern on the EtBr-free gel, so they don’t appear to be of different topological states.

→ We revised the indicated sentences to avoid potential errors and overinterpretation. We thank Reviewer 2 for his/her time and help to improve the manuscript.

REVIEWERS' COMMENTS

Reviewer #2 (Remarks to the Author):

Thank the authors for the responses! Only one concern left. In the relaxation assay of Fig. 5F, 200 ng of ERK proteins and 500 ng of plasmid were used in one reaction, which means that the concentration of ERK protein was 27-fold higher than that of the plasmid. If ERK really has DNase activity I will assume that it doesn't need such a high concentration to cut the plasmid.

REVIEWERS' COMMENTS

Reviewer #2 (Remarks to the Author):

Thank the authors for the responses! Only one concern left. In the relaxation assay of Fig. 5F, 200 ng of ERK proteins and 500 ng of plasmid were used in one reaction, which means that the concentration of ERK protein was 27-fold higher than that of the plasmid. If ERK really has DNase activity I will assume that it doesn't need such a high concentration to cut the plasmid.

→ We sincerely appreciate Reviewer 2 for acknowledging our efforts. To address Reviewer 2's concern, we titrated ERK1m to 0, 0.2, 2, 20, and 200 ng per a reaction in the DNA relaxation assay. The results have been newly included in the revised manuscript (**Supplementary Figure 7B**). In 20 and 200 ng ERK1m, the middle band could be observed, but not 0.2 and 2 ng reactions. As Reviewer 2 commented, 200 ng would make the DNA: protein ratio to be 1:27 and 20 ng would make 1:2.7. Although ERK1m doesn't appear to be robust, but mild, to relax/cut the supercoiled DNA, whether the seemingly weak reaction has significant/particular roles in the cell or not is unknown and will be important for us to understand in future. In addition, it is difficult to figure out how many ERK1m molecules in the 200 ng are active in the relaxation assays although our ERK1m protein appeared to be very active to phosphorylate TOP2B and ELK1 in the kinase assays. We look forward to understanding the interaction between ERK1m and DNA through rigorous experimentation in the future. Once more, we thank Reviewer 2.